# DEK promotes mammary hyperplasia and is associated with H3K27me3 epigenetic modifications

Megan E Johnstone[1],*, Ashley L Leck[1],*, Taylor E Lange[1,2], Katherine E Wilcher[1], Miranda S Shephard[1], Aditi Paranjpe[3], Sophia Schutte[1], Susanne I Wells[1,6], Ferdinand Kappes[4] , Nathan Salomonis[5,6], Lisa M Privette Vinnedge[1,6]

The DEK chromatin remodeling protein has oncogenic functions in breast cancers, but its functional role in normal mammary gland epithelium has remained unexplored. We developed two novel genetically engineered mouse models to study the role of Dek in normal mammary gland biology in vivo. Mammary gland-specific Dek transgenic mice developed hyperplasia and had a transcriptional profile that revealed increased expression of cell cycle, mammary stem/progenitor, and lactation-associated genes. Conversely, Dek knockout mice exhibited mammary gland functional defects resulting in dramatically reduced pup survival. Analysis of previously published scRNA-sequencing of mouse mammary glands revealed that *Dek* is most highly expressed in mammary stem cells and alveolar progenitor cells, supporting the observed phenotypes. Mechanistically, we discovered that Dek is a modifier of Ezh2 methyltransferase activity, up-regulating the levels of histone H3K27me3 to control gene transcription. Combined, this is the first report to show that Dek promotes proliferation of mammary epithelial cells via transcriptional deregulation of cell cycle genes, potentially via epigenetic mechanisms, in vivo.

## Introduction

Developmental pathways that maintain stem and progenitor cell pools, or promote cellular proliferation and migration during embryogenesis, are frequently reactivated and deregulated later in life during tumorigenesis (1). As an example, breast cancer is often characterized, in part, by the dysregulation of cell cycle (i.e., E2F and CDK4/6), DNA repair (i.e., BRCA1 and p53), and epigenetic modifying (i.e., EZH2) proteins (2, 3, 4, 5). In addition, chromatin remodeling and

epigenetic modification pathways that are important for controlling cellular differentiation are also deregulated during tumorigenesis (6, 7).

There is a substantial body of work on the role of the human DEK protein in solid and hematological malignancies; however, its roles(s) in normal development are poorly understood. DEK is a chromatin remodeling protein and histone H3 chaperone (8, 9, 10). In *Drosophila*, transgenic models over-expressing Dek in the eye resulted in a rough eye phenotype due to caspase-mediated apoptosis. This correlated with hypoacetylation of lysine residues in histones H3 and H4, including H3K27, H3K9, and H4K5, resulting in epigenetic silencing of the anti-apoptotic gene *bcl-2* (11). In tissue-specific knockout mouse models, Dek loss impaired the self-renewing ability of hematopoietic stem cells by limiting quiescence and accelerating mitochondrial metabolism. This led to decreased bone marrow cellularity and loss of hematopoietic stem and progenitor cells that was associated with increased H3K27ac epigenetic marks. This resulted in the transcriptional deregulation of several genes related to proliferation and metabolism, including up-regulated *Akt1/2*, *Ccnb1/2* (cyclin B), *Cdkn1a* (p21), and *Cdkn1b* (p27). The role of Dek in directing histone acetylation was found to occur through the recruitment of co-repressor NCoR1 and inhibition of histone acetyltransferases p300 and PCAF (12). In addition, Dek deficient primary mouse neurons cultured in vitro demonstrated an increase in acetylation of lysine 36 of histone H3 (H3K36ac) (13). These reports indicate that Dek suppresses histone H3 acetylation, leaving these residues available for methylation. Indeed, Dek was previously shown in *Drosophila* to support histone H3 trimethylation on residue lysine 9 (H3K9me3) (14). However, mechanisms whereby DEK promotes histone H3 methylation, and biological processes wherein this is important, remain unexplored. Histone methylation, particularly on histone H3, is both dynamic and tightly regulated during development and is important for establishing cell lineages, body patterning, and the development of specific organs. Histone methylation is accomplished by a family of methyltransferases, such as EZH2

[1]Division of Oncology, Cancer and Blood Diseases Institute, Cincinnati Children's Hospital Medical Center, Cincinnati, OH, USA [2]Department of Cancer Biology, University of Cincinnati College of Medicine, Cincinnati, OH, USA [3]Information Services for Research, Cincinnati Children's Hospital Medical Center, Cincinnati, OH, USA [4]Division of Natural and Applied Sciences, Duke Kunshan University, Kunshan, Jiangsu, China [5]Division of Biomedical Informatics, Cincinnati Children's Hospital Medical Center, Cincinnati, OH, USA [6]Department of Pediatrics, University of Cincinnati College of Medicine, Cincinnati, OH, USA

Correspondence: Lisa.Privette@cchmc.org
Megan E Johnstone's present address is Department of Internal Medicine, University of Cincinnati College of Medicine, Cincinnati, OH, USA
Katherine E Wilcher's present address is Boonshoft School of Medicine, Wright State University, Fairborn, OH, USA
*Megan E Johnstone and Ashley L Leck contributed equally to this work

of the polycomb repressive complex 2 (PRC2), and can be reversed by histone demethylases (15).

We, and others, have reported that DEK mRNA and protein are over-expressed in many human solid cancers, including 62–92% of breast cancers (16, 17, 18, 19). DEK has no known enzymatic functions but is a chromatin-modifying phosphoprotein with molecular activities in DNA repair, mRNA transcription, splicing, and histone modification (20, 21, 22, 23, 24, 25, 26). Recently, DEK was found to bind to the nucleosome in a manner similar to linker histones, as either a monomer or dimer, with the N-terminal tail embedded in the histone core, interacting with the octamer surface. The central SAP domain and C-terminus sit on top of the nucleosome, binding to the dyad and linker DNAs, bending the linker DNA to compact chromatin (27). We previously reported in vitro studies using MCF10A immortalized human mammary epithelial cells where we demonstrated that DEK over-expression promotes cellular proliferation and invasion in 2D cultures and 3D organoid models (17, 28). In 3D organotypic models of human skin, DEK over-expression promoted hyperplasia as evidenced by epidermal thickening and increased expression of proliferation-associated markers like PCNA (29). A squamous epithelium-specific *Dek* transgenic mouse model indicated that Dek over-expression did not promote hyperplasia in control conditions but did increase esophageal tumor formation in a 4NQO chemical carcinogenesis model (30). However, this differed from a *Drosophila* transgenic model, where Dek over-expression promoted apoptosis (11) and in the mouse hematopoietic system, where Dek loss, not over-expression, promoted cell cycle entry and suppressed quiescence (12, 31). Therefore, the role of Dek in promoting, or inhibiting, cellular proliferation has not been consistent across model systems. Furthermore, the impact of deregulated Dek expression on normal mammary epithelium has not been previously reported.

We sought to define the transcriptional and functional consequences of Dek over-expression in murine mammary epithelium. To this end, we generated a novel doxycycline-regulated mammary epithelium specific Dek overexpression model, MMTV-tTA/BiLDek (Dek-OE) mice and analyzed transcriptomic consequences of increased Dek expression in virgin mammary glands by RNA-sequencing. We report that Dek over-expression is sufficient to cause mammary epithelial hyperplasia characterized by excessive proliferation and cell cycle deregulation, and promotes expression of proteins associated with the luminal alveolar cell identity and milk production. We also report a novel *Dek* conditional knockout (Dek-cKO) mouse model with evidence of a primary lactation deficiency phenotype. Importantly, we report that Dek promotes H3K27me3 histone trimethylation in mammary epithelial cells in vivo and in vitro and physically interacts with Ezh2 and other members of the PRC2 complex in human and mouse cells.

## Results

### Mammary gland specific Dek over-expression in mice

To study the effects of Dek over-expression on normal mammary epithelium, we created a novel, tetracycline (doxycycline) responsive transgenic mouse model. Mouse mammary tumor virus-tetracycline transactivator (MMTV-tTA) mice (32) were crossed with the Bi-L-Dek mouse (30) resulting in a MMTV-tTA/Bi-L-DEK (DEK-OE) mouse (Fig 1A). In this model, doxycycline repressed the transcription of the *Dek* transgene and the luciferase reporter via a bidirectional promotor containing the tetracycline response element. Representative genotyping for the *tTA*, luciferase, and *Dek* transgenes and the endogenous *Dek* gene is shown in Fig 1B. Expression of the luciferase reporter transgene in bitransgenic MMTV-tTA/Bi-L-DEK (Dek-OE) mice (right), with repression by doxycycline chow ("Control", left), was validated by in vivo imaging system (IVIS, Fig 1C). Thus, the following experiments compare MMTV-tTA/Bi-L-DEK bitransgenic mice on dox chow ("Controls") to mice consuming normal chow ("Dek-OE"). We detected approximately twofold higher Dek protein expression in whole mammary gland protein lysates and via immunohistochemical staining in tissues collected from Dek-OE virgin females in diestrus compared with Dox chow repressed ("control") MMTV-tTA/Bi-L-DEK mice (Fig 1D–G). This is similar to the degree of over-expression observed in other in vitro and in vivo models (30, 33). Since mammary tumorigenesis can be a long process and is observed primarily in women over 50 yr of age (34), we aged nulliparous and multiparous females to the human equivalent age, which is 15 mo (~60 wk) (35), to detect the impact of prolonged Dek over-expression on mammary epithelium. Spontaneous tumor development was not observed during aging for either parity condition (data not shown). We then performed whole mount analysis to examine gland morphology from virgin aged adult females. There were no differences in branching morphogenesis, as there were no statistically significant differences in the number of primary or secondary ductal branches at multiple ages. However, we noted substantial hyperplasia in inguinal mammary glands from aged virgin adult Dek-OE mice, compared with + dox controls, marked by increased epithelial density as quantified by Sholl analysis in Fig 1H and I. This increased epithelial density was limited to a lobular-alveolar compartment and not ducts (36).

### RNA sequencing reveals differential expression of cell cycle related genes

We performed bulk RNA sequencing on whole mammary tissue from two + dox control and two Dek-OE adult virgin females at 15 mo of age to discover molecular targets regulated by Dek over-expression and to reveal a gene signature that could help identify the expanded cell population(s) in hyperplastic glands. Principle component analysis showed that the Dek over-expression transcriptome is unique and distinct from control samples (Fig 2A). We plotted differentially expressed genes (DEGs) by volcano plot and found that 1,631 genes were up-regulated and only 340 genes were down-regulated (Fig 2B). *Dek* mRNA over-expression in samples used for RNA-Seq was confirmed by plotting the FPKM for all samples (Fig 2C).

DEGs were plotted as a heatmap and ontologies for biomarkers of cell populations were defined to help identify the expanded cell population driving Dek-induced hyperplasia. Several gene signatures related to epithelial stem and progenitor cells were found but, most notably, a specific ontology for "adult mammary gland luminal progenitor" was identified among up-regulated genes in samples from Dek-OE mice compared with controls (Fig 2D red asterisk).

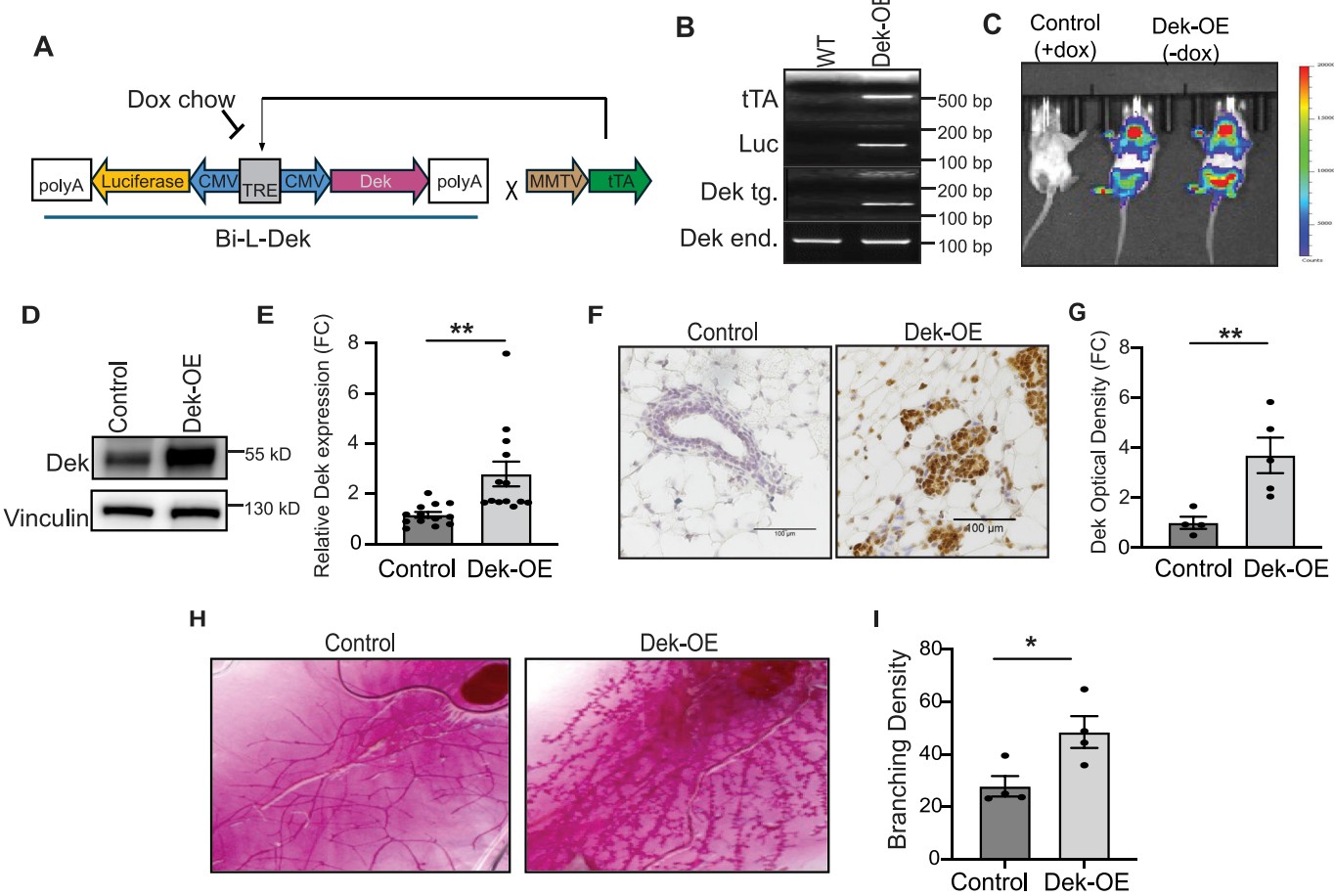

**Figure 1. Generation of a tetracycline-off regulated mammary gland specific overexpression of *Dek* transgenic murine model.**
**(A)** Transgenic construct of Bi-L-Dek transgenic mice with a tetracycline response element drives dual cytomegalovirus (CMV) promotors that bi-directionally transcribe luciferase and Dek. This transgene is transcribed by a tetracycline transactivator under the control of a murine mammary tumor virus promotor (MMTV-tTA), resulting in targeted mammary epithelium specific transgenic Dek over expression. **(A, B)** Representative genotyping reveals presence of transgenic constructs described in (A) and presence of endogenous (WT) Dek. **(C)** Examination of luciferase expression and tetracycline (doxycycline) repression in Dek-OE mice confirmed after intraperitoneal injection of luciferin and visualized with an in vivo imaging system. **(D)** Western blot analysis of Dek protein from whole lysed mammary gland. Vinculin served as a control for protein loading. **(E)** Western blot densitometry analysis reveals a 2-fold increase in DEK protein expression (N = 13/group, P = 0.0036). **(F)** Dek expression is elevated in Dek-OE mammary glands compared with + dox controls, as detected by immunohistochemistry. Scale bar = 100 μm. **(F, G)** Quantification of Dek immunohistochemistry from (F). Image J color deconvolution was used to measure staining intensity and is graphed as a fold-change (N = 4 (control) and N = 5 (Dek-OE) P = 0.0145 t test). **(H)** Representative whole mount images from +dox control and Dek-OE inguinal mammary glands. **(I)** Image J was used to perform Sholl analysis and quantify epithelial branching density of glands from 60-wk-old virgin females. (N = 4/group P = 0.0288). An unpaired t test was used for statistical analysis and data is presented as mean ± SEM.
Source data are available for this figure.

Indeed, we observed an up-regulation of genes associated with luminal alveolar progenitor mammary epithelial cells, including genes encoding milk proteins (Glycam1, Muc1, and caseins), luminal alveolar cell specific genes (Elf5, EpCam, Krt19, and EZH2), and stem/progenitor cell markers (CD44, Birc5/survivin, and Itga6/CD49f) (Fig 2B and E). We used gene set enrichment analysis (GSEA) to determine functional enrichment of DEGs. GSEA identified several cellular and molecular gene ontologies and KEGG pathways for up-regulated genes including cell cycle transition, DNA double-strand breast repair, chromatin remodeling, p53 signaling (NES = 1.69, FDR adjusted P = 0.00556), mammary gland development (including alveolus development), regulation of stem cell population maintenance, and prolactin signaling (NES = 1.75, FDR adjusted P = 0.00609) (Fig 2F) as well as up-regulation of genes needed for translation, nonsense-mediate decay

of mRNA, and telomere maintenance (Fig S1A). Down-regulated ontologies and pathways included oxidative phosphorylation, mitochondrial respirasome, NAD metabolic processes, sterol metabolic process, fatty acid biosynthesis, and cholesterol metabolic process (Fig S1B). Next, we used gProfiler for additional functional enrichment analysis of up-regulated genes in mammary glands from Dek-OE mice. This validated "cell cycle" and "metabolism of RNA/proteins" as central cellular processes deregulated in response to Dek over-expression compared with + dox controls (Fig S1C). We then used GO-Elite to identify transcription factors Yy1, Nfkb2, p53, and E2F as potentially responsible for regulating the expression of Dek-OE-induced DEGs (Fig S2A) whereas transcription factors associated with the expression of down-regulated genes were limited, but included Pparg, Myod1, and Srebf1 (Fig S3A). We also used GO-Elite to assess KEGG pathways

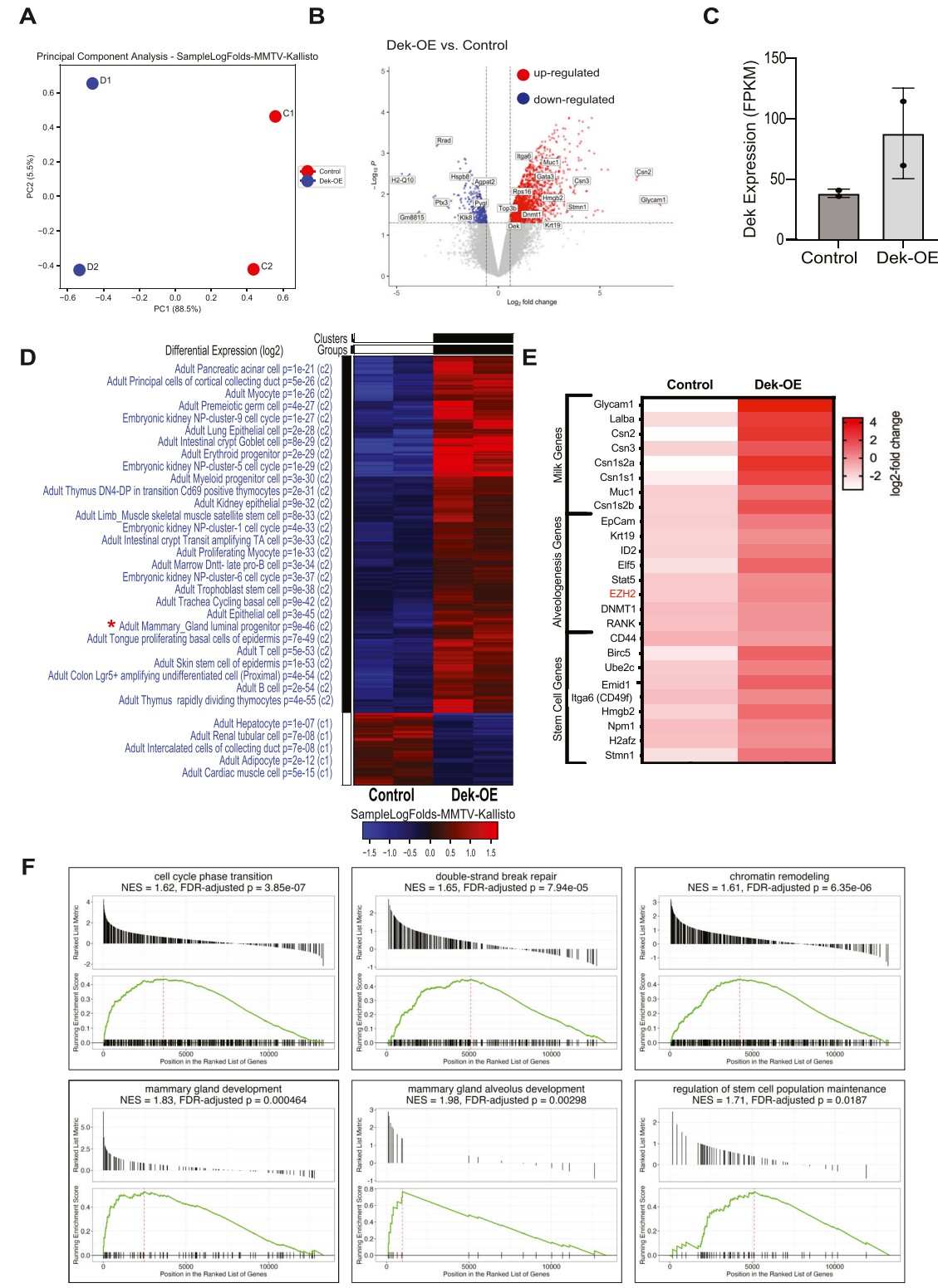

**Figure 2. RNA sequencing reveals differential expression of cell cycle regulatory signaling and markers of mammary luminal progenitor cells.**
**(A)** Principal component analysis reveals that Dek-OE samples (N = 2) are distinct and unique from control samples (N = 2). Samples were collected from 60-wk-old virgin females. **(B)** Volcano plot is used to visualize significant differentially up- and down regulated gene targets. Up-regulated targets are depicted in red and down-regulated targets in blue. **(C)** *Dek* mRNA levels of samples used for bulk RNA-Seq. **(D)** A heat map depicts the up- and down-regulated genes in Dek-OE mammary glands compared with + dox chow controls. Top hits of the most significant gene ontologies associated with the up- and down-regulated genes are listed on the left. * Note the ontology of "adult mammary gland luminal progenitor" $P = 9 \times 10^{-46}$. **(E)** A visual representation of a select set of genes strongly over-expressed in mammary glands from

associated with DEGs caused by Dek over-expression. For up-regulated genes, we identified p53 signaling, cell cycle, tight junctions and adherens junctions, ribosomes, and progesterone signaling (Fig S2B). For down-regulated genes, we identified KEGG pathways related to adipocytokine signaling and a small number of genes related to various metabolic pathways like glycolysis, citrate cycle, and pentose phosphate pathways (Fig S3B). The observed gene expression profile indicates a possible Dek dependent regulation of cellular processes relevant to mammary gland development, including cell cycle progression, metabolism, and progesterone signaling, that may promote hyperplasia in Dek-OE mammary epithelium.

Given the hyperplasia phenotype, and functional enrichment analyses identifying "cell cycle phase transition" to be deregulated with Dek over-expression, we validated the respective DEGs with a focus on cyclins, cyclin-dependent kinases, and cell cycle inhibitors. Immunohistochemical staining of 8-wk old inguinal glands from virgin females showed that expression levels of cell cycle inhibitors p21 and p27, which inhibit cyclin-CDK complexes, were significantly down-regulated with Dek over-expression (Fig 3A and B). Cyclin A and Cdk proteins -2, -4, and -6 were also found to be highly expressed in mammary epithelium from Dek-OE mice compared with + dox controls (Fig 3C–F). To test if the deregulation of cell cycle genes contributed to hyperplasia, we isolated primary mammary epithelial cells from control and Dek-OE juvenile mice and cultured them as 3D organoids. Indeed, Dek over-expressing organoids were nearly three times larger in volume compared with + dox controls. Importantly, in vitro we treated organoids with the CDK4/6 inhibitor palbociclib, an FDA-approved small molecule inhibitor used to treat estrogen receptor positive (ER+) breast cancer. We observed that the increased growth of Dek over-expressing organoids was dependent on the Dek-induced increase in CDK4/6, since palbociclib treatment resulted in smaller Dek over-expressing organoids that were comparable to organoids from +dox controls (Fig 3G and H). This effect was most notable by day 7 of organoid growth. The observed deregulation of cell cycle effectors was sufficient to create a pro-proliferative phenotype. As early as 6 wk of age, we detected an increased proportion of Ki67 positive proliferative cells in mammary epithelium in virgin Dek-OE females compared with + dox controls (Fig 3I and J). Of note, the association between *Dek* and a pro-proliferative gene signature was not limited to mouse mammary epithelium. Using the METABRIC dataset in The Cancer Genome Atlas, we determined that *DEK* mRNA expression was inversely correlated with *CDKN1A* (p21; Fig S4A) and positively correlated with *CCNA2* (cyclin A), *MKi67* (Ki67), and *PCNA* (Fig S4B–D). This suggests that transcriptionally mediated consequences of Dek over-expression in murine models may be relevant in humans.

## Dek promotes expression of genes associated with luminal alveolar cells

Dek over-expression correlated with a cellular ontology of "mammary gland luminal progenitor cells" and several up-regulated genes are

linked to milk protein production. To validate these transcriptomic profiles translated to increased protein expression, we performed western blot and immunohistochemical staining for luminal alveolar markers and milk proteins on whole mammary glands from 8-wk-old virgin females. As predicted, Dek-OE mammary glands produced higher levels of milk proteins like Csn2 (β-casein; Fig 4A and B) and Muc1 (Mucin1; Fig 4C and D), even in virgin glands, and higher expression of luminal alveolar cell markers such as cytokeratins-7 and -19 and EpCAM (Fig 4E and F). However, progesterone receptor, estrogen receptor alpha (ERα/*Esr1*) and HER2 (*Erbb2*) were unchanged in Dek-OE glands compared with controls, and there were no consistent differences that reached statistical significance in the expression of genes associated with hormone sensing luminal cell and basal cell populations (Fig S5A–C). This indicated that Dek function is limited in these populations and that Dek expression is likely downstream of, or unrelated to, PR, ERα, and HER2.

Luminal alveolar progenitor cells expand rapidly during pregnancy due to hormonal stimulation. We therefore investigated if pregnancy would impact gland morphology in Dek-OE glands compared with + dox controls. Indeed, whole mount analyses of mammary glands from 12.5 dpc pregnant mice demonstrated that Dek over-expression led to hyperplasia compared with + dox controls (Fig 4G and H). Since cellular expansion of the mammary gland is hormonally driven, and since *DEK* was previously published to be an ERα target gene in human cells (16), we also determined whether endogenous Dek expression in the mammary gland was dependent on ovarian hormones. Immunohistochemistry staining revealed that Dek expression in the mammary epithelium was reduced by nearly 50% within 1 wk of ovariectomy compared with intact controls (Fig 4I and J). This demonstrated that ovarian hormones, like progesterone and estrogen, partially promote Dek expression in mammary epithelium and that *Dek* is downstream of steroid hormone signaling, in agreement with our previous in vitro studies (16).

To assess which mammary epithelial cell subpopulations express the highest levels of *Dek*, we next analyzed *Dek* expression in a single cell murine mammary cell gene atlas, which was created by Saeki et al. by combining four single cell RNA sequencing datasets (37). *Dek* mRNA levels were highest in mammary stem cells (MaSC), luminal alveolar progenitors (LA-pro), and luminal hormone receptor positive progenitor (LH-pro) cells (Fig 5A). A subset of basal cells also expressed *Dek* whereas minimal expression was observed in mature, differentiated luminal cells. Importantly, these data support our bulk RNA-Seq data from Dek over-expressing mammary glands in that both datasets link *Dek* with luminal alveolar and stem/progenitor cell markers (see Fig 2). GSEA of the genes most strongly correlated with *Dek* expression in the LA-pro cluster identified cell proliferation ("E2F targets"), chromatin remodeling, and epigenetic regulation as the most enriched gene sets associated with *Dek* levels (Figs 5B, S6, and S7). The 25 genes that were most strongly correlated with *Dek* are shown in Fig 5C. Cell

---

Dek-OE mice compared with + dox controls that are indicative of an increase in luminal alveolar progenitor cells. Genes associated with milk protein production, alveologenesis, and stem cells are shown. **(F)** Gene set enrichment analysis (GSEA) plots for genes up-regulated in glands from Dek-OE mice, depict enrichmenet for genes related to cell cycle phase transition, DNA double-strand break repair, chromatin remodeling, mammary gland development, mammary gland alveolus development, and regulation of stem cell population maintenance.

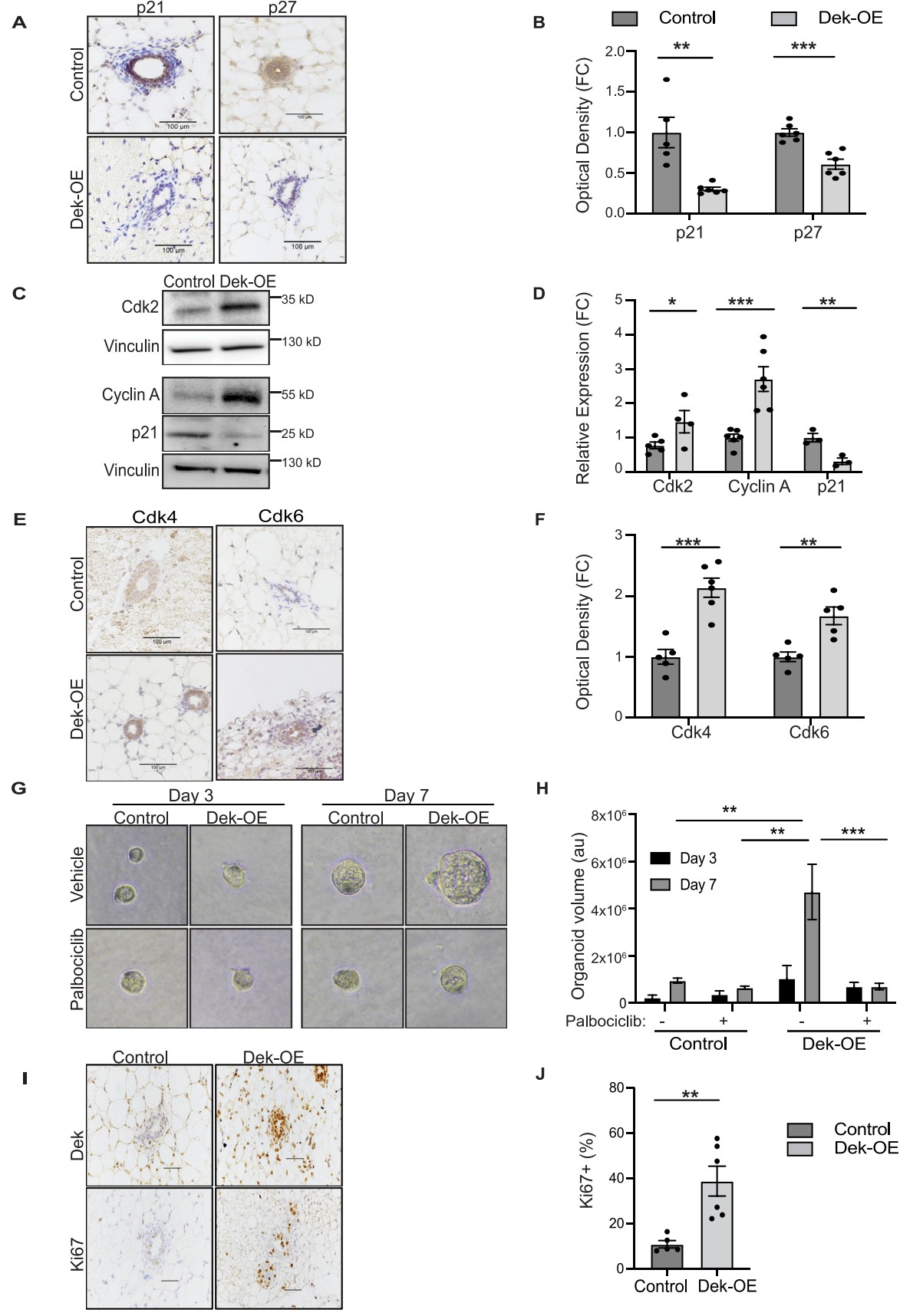

clusters expressing the correlated epigenetic factor and histone methyltransferase, *Ezh2*, the proliferation marker *Pcna*, and the *E2F1* transcription factor that regulates the G1/S cell cycle transition, also overlapped with those expressing high levels of *Dek* (Fig 5D). Importantly, we noticed that two components of the PRC2 complex, *Ezh2* and *Rbpp4*, were strongly correlated with *Dek* expression and were highly expressed in the same three cell clusters as *Dek*: MaSC, LH-pro, and LA-pro cells (Fig 5E). Another PRC2 complex member, *Suz12* (correlation coefficient = 0.24), was also correlated with *Dek*, although it was not in the list of top 25 most correlated genes. The PRC2 complex uses the histone methyltransferase activity of Ezh2 to trimethylate histone H3 on lysine 27 (H3K27me3). Since Dek is known to function as a histone H3 chaperone in other systems, we tested whether Dek is a co-factor regulating histone H3 epigenetic post-translational modifications.

### Dek increases H3K27 trimethylation in murine and human cells

Dek has previously been shown to repress histone H3 acetylation of lysine residues, which leaves these sites available for methylation (11, 14, 38). We quantified *Ezh2* mRNA expression in the bulk RNA-Seq samples. A 4.2-fold increase of *Ezh2* was observed in mammary gland tissue from Dek-OE mice compared with + dox controls (Figs 2E and 6A). A significant increase in H3K27me3 levels, relative to total histone H3, was observed by Western blotting of mammary gland lysates from virgin females (Fig 6B). We validated Ezh2 and H3K27me3 up-regulation specifically in mouse mammary epithelial cells by immunohistochemistry (Fig 6C). To test if DEK over-expression correlated with EZH2 and H3K27me3 in human cells, DEK was over-expressed or knocked-down in human MCF10A immortalized mammary epithelial cells. We observed a positive correlation, with H3K27me3 levels increased upon dox-induced DEK over-expression and decreased in DEK-shRNA knockdown cells (Fig 6D). We also validated this using primary human mammary epithelial cells. Cells were transduced with DEK shRNA lentivirus to target DEK expression (Fig S8A). Loss of DEK resulted in increased p27 and decreased Survivin expression (Fig S8A). DEK deficient cells ceased to proliferate, with cell numbers not increasing over 4 d of culture (Fig S8B) and having a senescent-like morphology (Fig S8C). Likewise, DEK shRNA primary HMECs showed a striking loss of

H3K27me3 levels and few Ki67+ proliferating cells as determined by immunofluorescence in 2D and 3D organoid cultures (Fig S8C and D). To probe a potential interaction between DEK and components of the PRC2 complex, a GFP-trap assay was performed with GFP-tagged DEK in HEK293 cells. GFP-DEK interacted with EZH2, RBBP4, and EED of the PRC2 complex as well as histone H3 (Figs 6E and S5D). CK2 was used as a positive control for a DEK interacting protein (39, 40). Notably, endogenous DEK was also pulled down in the GFP-trap assay, confirming that GFP-DEK can still multimerize with untagged DEK, as previously reported (Fig 6E) (41). In additional, endogenous DEK interacted with endogenous EZH2 and SUZ12 in MCF10A immortalized human mammary epithelial cells by immunoprecipitation (Fig 6F). To determine if DEK and EZH2 expression correlate in other models, we quantified their expression in breast cancer datasets. DEK and EZH2 protein (Fig 6G) and mRNA (Fig 6H) expression levels were, indeed, strongly positively correlated in human breast cancers. Interestingly, *DEK* and *EZH2* mRNA levels tended to be higher in estrogen receptor (ER) negative breast cancers (orange dots, Fig 6H). Next, we wanted to determine if Ezh2 methyltransferase activity and increased H3K27me3 levels contributed to the hyperplasia observed with Dek over-expression. Primary mouse mammary epithelial cell organoids from +dox control and Dek-OE mice were cultured as organoids and treated for 7 d with the Ezh2 inhibitor GSK-126. As previously demonstrated, organoids from Dek-OE mammary glands were significantly larger than those from +dox controls, but this increased volume was prevented with Ezh2 inhibition, suggesting Dek-induced hyperplasia may be dependent on Ezh2 methyltransferase activity (Figs 6I and S5E). In summary, DEK promotes PRC2 complex-mediated deposition of H3K27me3 in human and murine mammary epithelial cells and that DEK/H3K27me3 loss correlates with suppressed proliferation.

We next created a Dek-floxed mouse model and bred it to the CMV-Cre mouse to create a constitutive Dek knockout ("Dek cKO" Fig 7A). Dek protein loss was confirmed by Western blot analysis of whole mammary tissue with a decrease noted for cytokeratin 7 expression, suggesting a decrease in the luminal cell population within the mammary gland (Fig 7B). Interestingly, at the time of weaning (21–28 d post-birth), litter sizes of Dek heterozygous and knockout dams were significantly smaller than litters of both WT

**Figure 3.  Dek-OE mice display a mitogenic phenotype associated with increased expression of cyclin-CDK complex proteins.**
**(A)** Dek-OE mammary epithelium has decreased expression of cell cycle inhibitor proteins p21 and p27 as detected by immunohistochemistry. Scale bar = 100 mm. **(B)** Quantification of immunohistochemical staining of p21 and p27 from subpanel A using Image J color deconvolution and represented as fold change. For p21, N = 5/group (*P* = 0.0027) and for p27 N = 6/group (*P* = 0.0005). **(C)** Representative western blot images showing decreased p21 protein expression and increased expression of Cdk2 and cyclin A in mammary glands form Dek-OE mice compared with + dox controls. Vinculin served as protein loading controls. **(D)** Quantification of Western blots in shown in subpanel C using densitometry analysis. For Cdk2, N = 5/group (*P* = 0.05), for cyclin A, N = 6/group (*P* = 0.0010), and for p21, N = 3/group (*P* = 0.011). **(E)** Mammary glands over-expressing Dek have higher expression of Cdk4 and Cdk6 compared with control glands as detected by immunohistochemistry. Scale bar = 100 $\mu$m. **(E, F)** Quantification of immunohistochemical staining of Cdk4 and Cdk6 shown in subpanel (E) using Image J color deconvolution to measure optical density, graphed as fold-change compared with + dox control samples. n = 5/group. For Cdk4, *P* = 0.0004 and for Cdk6 *P* = 0.004. **(G)** Palbociclib, a Cdk4/6 small molecule inhibitor, prevents Dek-induced hyperplasia. All organoids were imaged with a 20x objective lens and cropped to the same size. **(H)** Quantification of organoid volume from subpanel G using measurements obtained with Image J. Organoid volume is presented as mean ± SEM and statistical significance determined using a two-way ANOVA (group x treatment) for each day. There were no significant differences between any samples on day 3. Two-way ANOVA revealed a time × group interaction (F[3,10] = 8.475, *P* = 0.0042) and statistically significant values were only identified for the DEK-OE + palbociclib group compared with control groups on day 7. Organoids were cultured from primary mammary tissue isolated from N = 3 mice (control) and N = 5 mice (Dek-OE). **P < 0.01, ***P < 0.001. **(I)** Dek-OE mammary glands have significantly more proliferating mammary epithelial cells as determined by Ki67 immunohistochemical staining. Scale bar represents 50 $\mu$m. **(I, J)** Quantification of Ki67 staining form subpanel (I), with the percentage of Ki67+ nuclei within mammary epithelium shown. (N = 5/control and N = 6/Dek-OE, *P* = 0.0045). Virgin mice were collected at 6–8 wk of age for each experiment. For each bar graph, each mouse is depicted by individual dots. Graphs are presented as mean ± SEM and statistical significance is determined by an unpaired *t* test unless otherwise noted. *P < 0.05, **P < 0.01, ***P < 0.001, ns = not significant.
Source data are available for this figure.

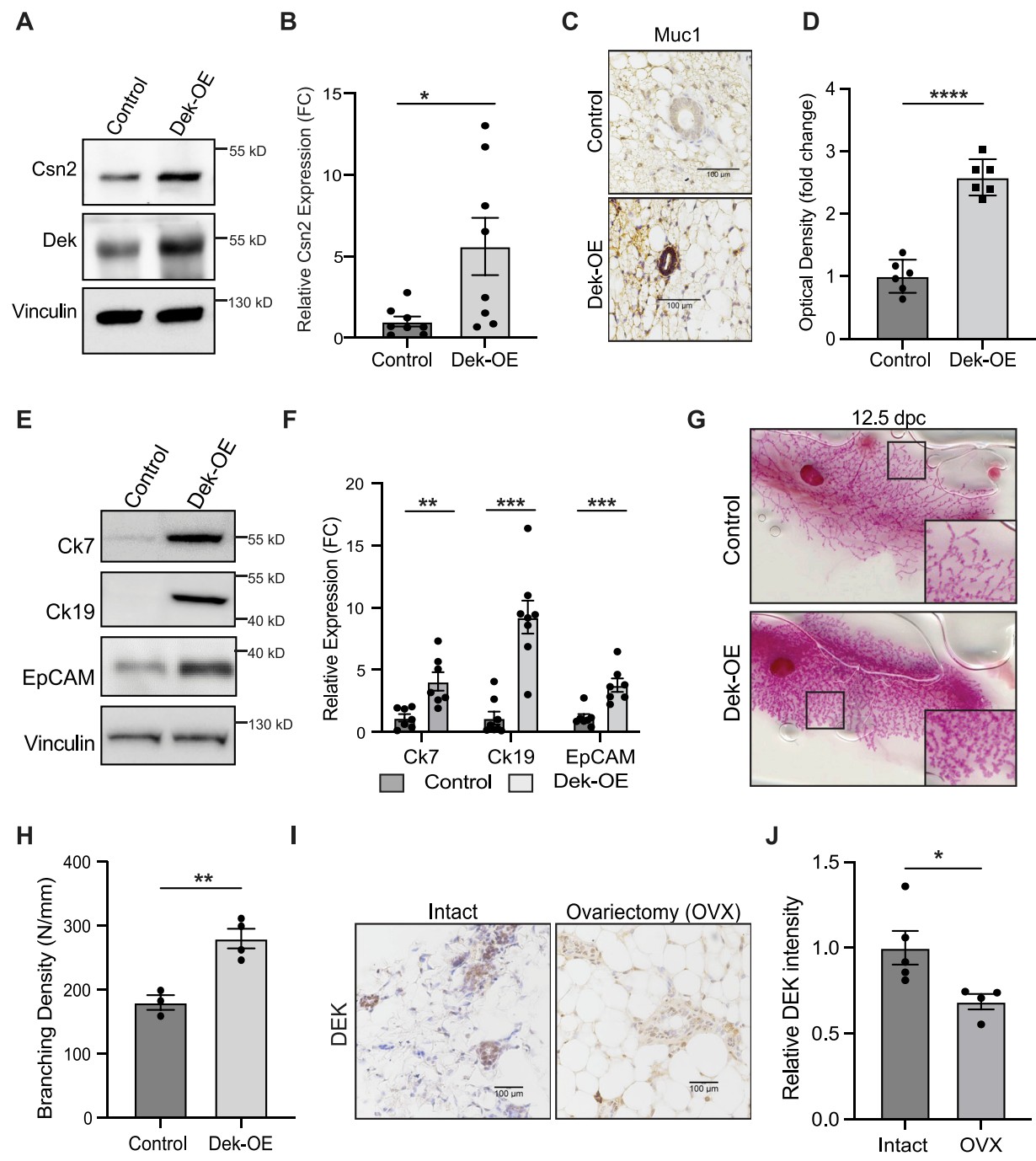

**Figure 4. Dek overexpression expands the luminal-alveolar cellular compartment of the mammary epithelium.**
Dek over-expression correlates with expression of luminal alveolar markers and epithelial expression during pregnancy. **(A)** Mammary glands from Dek-OE mice express more Csn2 (β-casein) than controls as determined by Western blotting of whole gland tissue. Tissue was collected from 8-wk-old virgin females. **(A, B)** Quantification of Western blot images from (A) using densitometry. N = 8/group (P = 0.0216). **(C)** Dek-OE mammary epithelium expresses significantly more Muc1 (Mucin1) compared with + dox controls as detected by immunohistochemical staining. Tissues were collected from 8-wk-old virgin females. Scale bar = 100 μm. **(C, D)** Quantification of staining intensity for Muc1 immunohistochemistry shown in (C). N = 6/group, P < 0.0001. **(E)** Luminal alveolar markers cytokeratins -7 and -19 and EpCAM are highly expressed in whole mammary glands from Dek-OE mice compared with + dox controls, as detected by Western blotting of whole tissue from 8-wk-old virgin female mice. **(E, F)** Densitometry quantification of Western blots from (E). For Ck7, N = 7/group (P = 0.0035). For Ck19, N = 8/group (P < 0.0001). For EpCAM, N = 7/group (P = 0.0010). **(G)** Whole mount images reveal that Dek over-expression results in increased expansion of mammary epithelium during pregnancy at 12.5 d post-coitum. Specifically, the Dek-OE expansion is in the lobuloalveolar compartment. **(H)** Quantification of whole mounts from (G) using Sholl analysis via Image J. N = 3 (control) and N = 4 (Dek-OE), P = 0.0047. **(I)** Dek expression, detected by immunohistochemistry, in mammary epithelial cells is decreased 1 wk after ovariectomy (OVX, n = 4) compared with intact (non-OVX, n = 5) female mice. Scale bar = 100 μm. **(I, J)** Quantification of staining intensity from (I) using Image J color deconvolution to detect DEK staining in epithelial cells. Graphs are presented as mean ± SEM and statistical significance is determined by an unpaired t test. *P < 0.05, **P < 0.01, ***P < 0.001 ns = not significant. Source data are available for this figure.

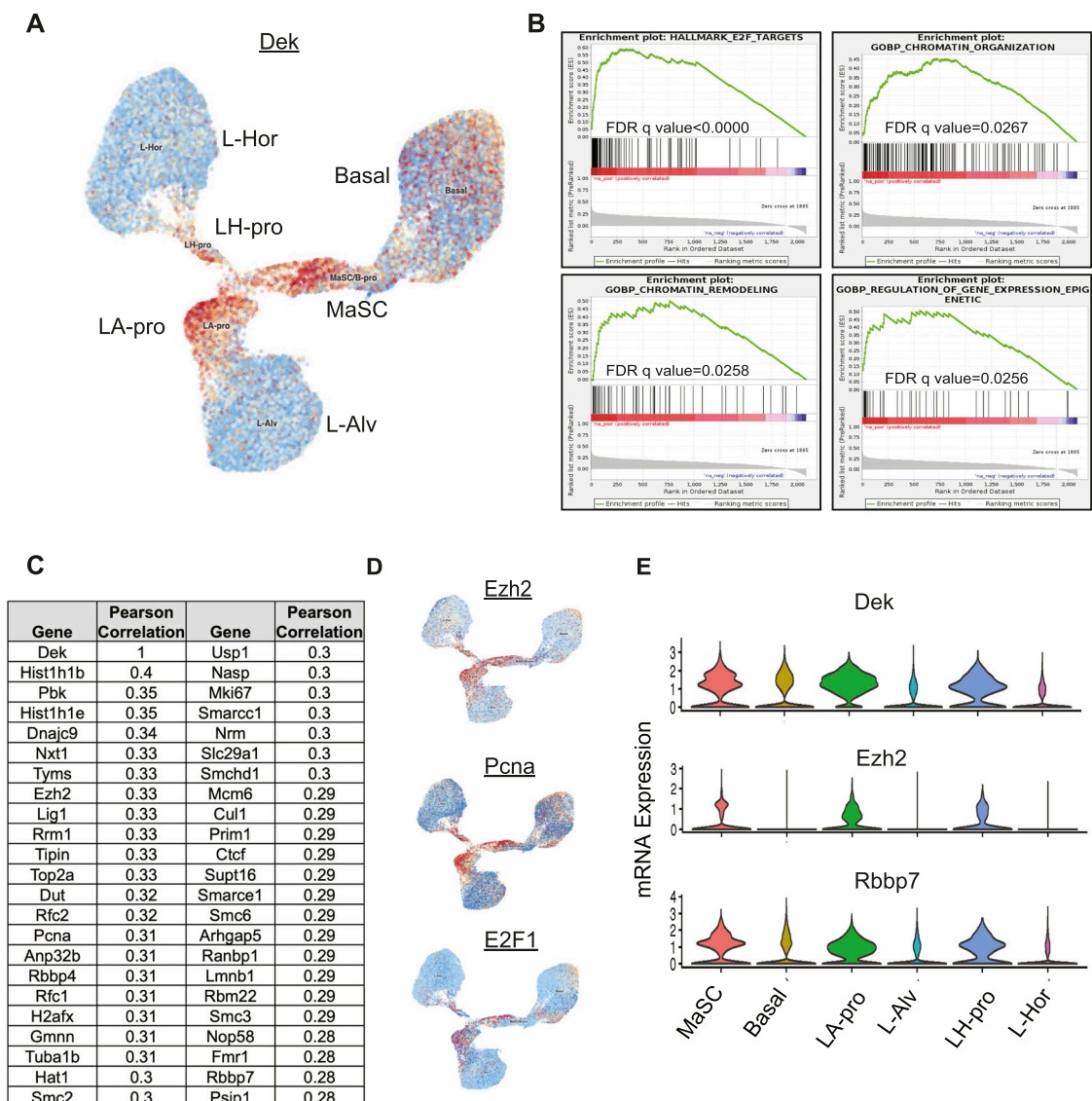

**Figure 5. Dek is highly expressed in mammary stem cells and luminal progenitor cells and is co-expressed with genes associated with proliferation and chromatin remodeling.**
**(A)** UMAP plot depicting Dek expression (red) in single cell RNA-sequencing data from mouse mammary gland previously published by reference 37. The populations are mammary stem cells, luminal alveolar progenitor cells (LA-pro), luminal hormone receptor positive progenitors (LH-pro), basal cells, mature luminal alveolar cells (L-Alv), and mature luminal hormone receptor positive cells (L-Hor). **(B)** Gene set enrichment analysis plots for genes co-expressed with Dek in the luminal alveolar progenitor cellular compartment. Plots depict E2F Targets, Chromatin Organization, Chromatin Remodeling, and Regulation of Gene Expresion-Epigenetic ontologies. **(C)** A list of the top genes co-expressed with Dek in the luminal alveolar progenitor cell population. **(A, D)** UMAP plots for gene expression of *Ezh2*, and proliferation-associated genes *Pcna* and *E2F1*, overlap with the expression of Dek in (A). **(C)** All three genes are in the list shown in (C). **(E)** *Dek*, *Ezh2*, and *Rbbp7* mRNA expression levels in each mammary cell population are depicted as a violin plot. All three genes are most highly expressed in MaSC, LA-pro, and LH-pro populations.

(Dek^+/+) and CMV-Cre^−/Dek^fl/fl control dams (Fig 7C). Survival analysis revealed that at least half the pups born to Dek-deficient females died within 24 h of birth (Fig 7D). Importantly, pup survival was not dependent on Dek genotype of the pups (data not shown) but was dependent on the genotype of the mothers. Pups born to Dek-deficient dams were dehydrated and lacked a milk spot, suggesting insufficient milk uptake (Fig 7E). Histological evaluation of mammary glands collected from lactating females suggested that the loss of Dek resulted in impaired mammary gland expansion during alveologenesis, as determined by H&E staining that showed

filled adipocytes and significantly less epithelium in Dek-deficient mammary glands, and insufficient milk production as shown by decreased Muc1 stained milk (Fig S9). Furthermore, whole mount visualization revealed that mammary glands from Dek cKO adult virgin females were smaller and less complex, with reduced branching density (Fig 7F). This hypoplasia correlated with increased expression of cell cycle inhibitor p21 (Fig 7H). Finally, we investigated if Dek loss impacted H3K27me3 levels, like Dek loss in human MCF10A cells (Fig 6D). Mammary glands from virgin adult Dek knockout mice harbored significantly less H3K27me3 than controls

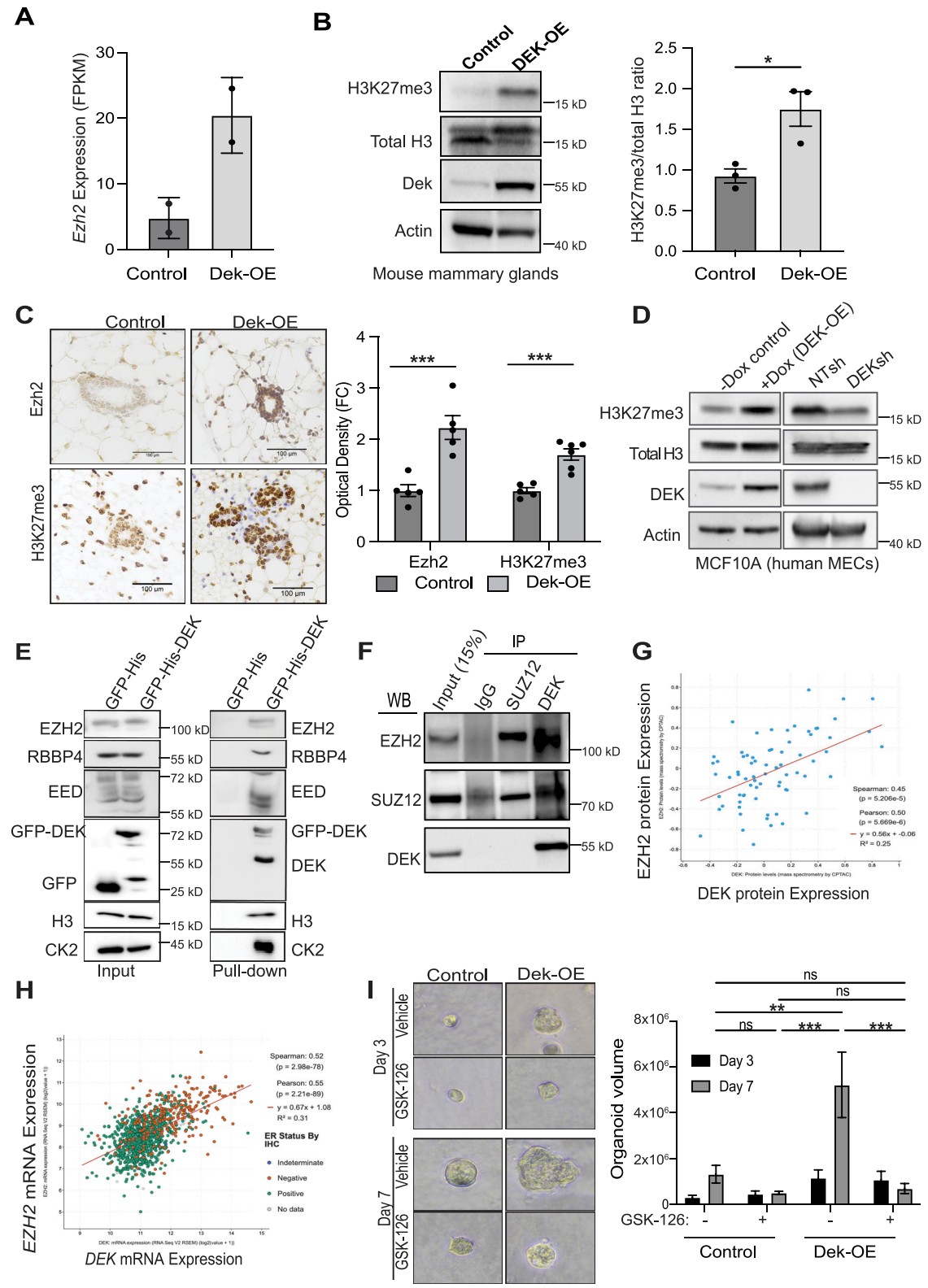

**Figure 6. Dek interacts with the PRC2 complex to promote H3K27me3 epigenetic modifications in mouse and human cells.**
**(A)** *Ezh2* gene expression from bulk RNA-Seq of +dox control versus -dox Dek over-expressing mouse mammary glands from MMTV-tTA/Bi-L-Dek mice, as described in Fig 2 (n = 2/group, 60-wk old virgin females). **(B)** Mammary glands from Dek-OE mice have increased levels of histone H3 trimethylation on residue lysine 27 (H3K27me3) compared with + dox controls, as shown by western blot. Densitometry quantification is graphed to the right. N = 3 mice/group. Graphs are presented as mean ± SEM and statistical significance is determined by an unpaired *t* test. *$P < 0.05$, **$P < 0.01$, ***$P < 0.001$ ns = not significant. **(C)** H3K27me3 and EZH2 protein levels are increased in

by Western blot analysis (Fig 7G) and immunohistochemistry (Fig 7H), that was accompanied by decreased levels of Ezh2 (Fig 7H). Combined, this indicates that mammary glands from Dek deficient mice are poorly functional and have impaired epigenetic remodeling, as evidence by decreased H3K27me3 levels. We conclude that Dek expression supports the proliferation of luminal alveolar progenitor cells and is associated with epigenetic reprogramming by stimulating EZH2-mediated histone H3 trimethylation at lysine 27.

# Discussion

Here, we are the first to describe a novel murine mammary gland model of temporal, tissue-specific Dek over-expression with transcriptomic profiling. We find that greater than twofold up-regulation of Dek promotes epithelial hyperplasia originating from alveolar buds, not ducts, that is characterized by cell cycle deregulation. Many studies have shown DEK to be highly expressed in proliferating human cells, as defined by co-expression between DEK and Ki67 or BrdU + cells, but it was not clear if DEK expression could promote the cell cycle. Here, we find adult Dek-OE mice display significantly more epithelium than their + dox control counterparts, coinciding with an imbalance of cyclin A, CDK2, p21, and p27 that leans toward a pro-proliferative signaling mechanism. The cumulative life-time effect of discrepancies in cell cycle signaling resulted in hyperplastic mammary epithelium in Dek transgenic mice. Previously, the molecular mechanism(s) by which DEK/Dek promoted proliferation were not well defined. We note that cell cycle deregulation appears to begin from the early stages of cell cycle control, via down-regulation of cyclin/CDK inhibitors p21 and p27. This suggests that Dek expression may be a key component of exit from quiescence and entry into the G1 phase of the cell cycle and cell cycle progression. Indeed, DEK as a mediator of exit from quiescence was previously reported in breast cancer stem cells, hematopoietic multipotent progenitors, muscle satellite (progenitor) cells and *Artemia* crustaceans (25, 42, 43, 44, 47). Importantly, DEGs in Dek over-expressing mammary glands were associated with both the cell cycle and p53 (Fig 2F). This aligns with previous reports that DEK silencing by RNAi in glioblastomas and HeLa cells increases p53 up-regulation and subsequent *CDKN1A*/p21 expression whereas exogenous DEK silences *TP53* (p53) expression

(44, 45, 46). p53 is a key regulator of cell cycle control and quiescence, including maintaining quiescence in epithelial progenitor cells through the regulation of *CDKN1A*/p21 expression.

Transcriptional control of *Cdkn1a* is positively regulated by p53 but both *Cdkn1a* and *Cdkn1b* are transcriptionally repressed via H3K27me3-mediated epigenetic silencing of their promoters (2, 48, 49). Histone H3 trimethylation on lysine 27 (H3K27me3) is accomplished through the methyltransferase activity of Ezh2 and the PRC2 complex. Here, we report for the first time that Dek is co-expressed with Ezh2 in murine and human mammary tissues, promotes the total deposition of the H3K27me3 epigenetic mark in mammary epithelial cells, and physically interacts with PRC2 complex members EED, RBBP4, and EZH2 in human cells. It will be necessary, in the future, to perform ChIP-Seq experiments to determine if the localization of these H3K27me3 marks facilitated by DEK are deposited at functionally relevant loci within the genome. Whereas this manuscript was in revision, a recent report demonstrated that Dek stimulates the PRC2 complex to promote the deposition of the H3K27me3 epigenetic mark on nucleosomes using a cell-free system. Furthermore, ChIP-Seq analysis of neural progenitor cells from the brains of E11.5 mouse embryos revealed that ~60% of Dek peaks overlapped with H3K27me3 peaks near transcription start sites (27). Combined, this indicates that Dek facilitates the regulation of gene transcription in progenitor cells, across multiple tissues, potentially through the PRC2 complex-mediated trimethylation of histone H3. However, it is important to note that epigenetic and transcriptional regulation of cell cycle genes by Dek/Ezh2/H3K27me3 in the mammary gland could be direct or indirect. Direct regulation might involve Dek cooperating with Ezh2 and the PRC2 complex to deposit the H3K27me3 mark directly on the promoters of quiescence-associated genes like *Cdkn1a* and *Cdkn1b*, leading to their transcriptional silencing and persistent proliferation via Cdk4/6 activity to drive hyperplasia. Indirect regulation might relate to prior work wherein DEK has been linked to several mitogenic signaling pathways including AKT/mTOR and Wnt signaling, which could subsequently up-regulate cell cycle gene expression to promote a pro-proliferative state (12, 17, 50, 51). Future studies will seek to define direct or indirect modes of regulation by defining functionally relevant Dek/Ezh2/H3K27me3-dependent molecules and pathways.

---

Dek-OE mammary epithelium from 8-wk old virgin female mice, compared with controls, as determined by immunohistochemistry. Staining intensity, quantified as optical density using Image J, is graphed to the right with N = 5 mice/group and statistical significance determined using an unpaired *t* test. **(D)** H3K27me3 levels positively correlate with DEK expression in human MCF10A cells either over-expressing DEK (left) or with DEK knockdown with shRNA (right) as determined by western blot of whole cell lysates. DEK over-expression was accomplished with a dox-inducible pTRIPZ vector whereas DEK knockdown was accomplished with a pLKO.1 shRNA vector. **(E)** Immunoblotting of GFP-trap nuclear lysates identify proteins that interact with GFP-DEK. Interacting proteins include key members of the PRC2 complex, EZH2, RBBP4, and EED, as well as endogenous DEK, histone H3, and the known DEK-interacting positive control of CK2. Input lysates, pre-incubation with GFP trap beads, are shown adjacent to the pull-down results. **(D, F)** DEK interacts with PRC2 complex members EZH2 and SUZ12 as determined by immunoprecipitation using whole cell lysates from MCF10A cells with dox-induced DEK overexpression using the same pTRIPZ construct shown in (D). **(G)** EZH2 and DEK protein levels show a strong positive correlation in human primary invasive breast cancers. Pearson correlation = 0.50, $P = 5.669 \times 10^{-6}$. **(H)** *EZH2* and *DEK* mRNA levels, detected by RNA-Seq, show a strong positive correlation in human primary breast cancers. Pearson correlation = 0.55, $P = 2.21 \times 10^{-89}$. Estrogen receptor (ER) negative samples, as determined by immunohistochemistry, are shown as orange circles whereas ER positive samples are in green. Blue and gray dots are samples with indeterminate or no data, respectively. Data for (G, H) are from the TCGA Firehose Legacy dataset for breast invasive carcinoma, accessed using www.cbioportal.org, and graphed using a log scale. Trend lines are shown in red. **(I)** EZH2 inhibition with GSK-126 limits hyperplasia in Dek-OE 3D organoids with no significant impact on control organoids. All organoids were imaged with a 20x objective lens and cropped to the same size. Organoid volume is graphed on the right and presented as mean ± SEM and statistical significance determined using a two-way ANOVA (genotype x treatment) for each day. There were no significant differences between any samples on day 3. Statistically significant differences were detected between groups on day 7. N = 3 mice (control) and N = 5 mice (Dek-OE) and primary tissue was collected from virgin adult females 8–12 wk old. Multiple organoids were quantified per donor. *$P < 0.05$, **$P < 0.01$, ***$P < 0.001$, ns = not significant.
Source data are available for this figure.

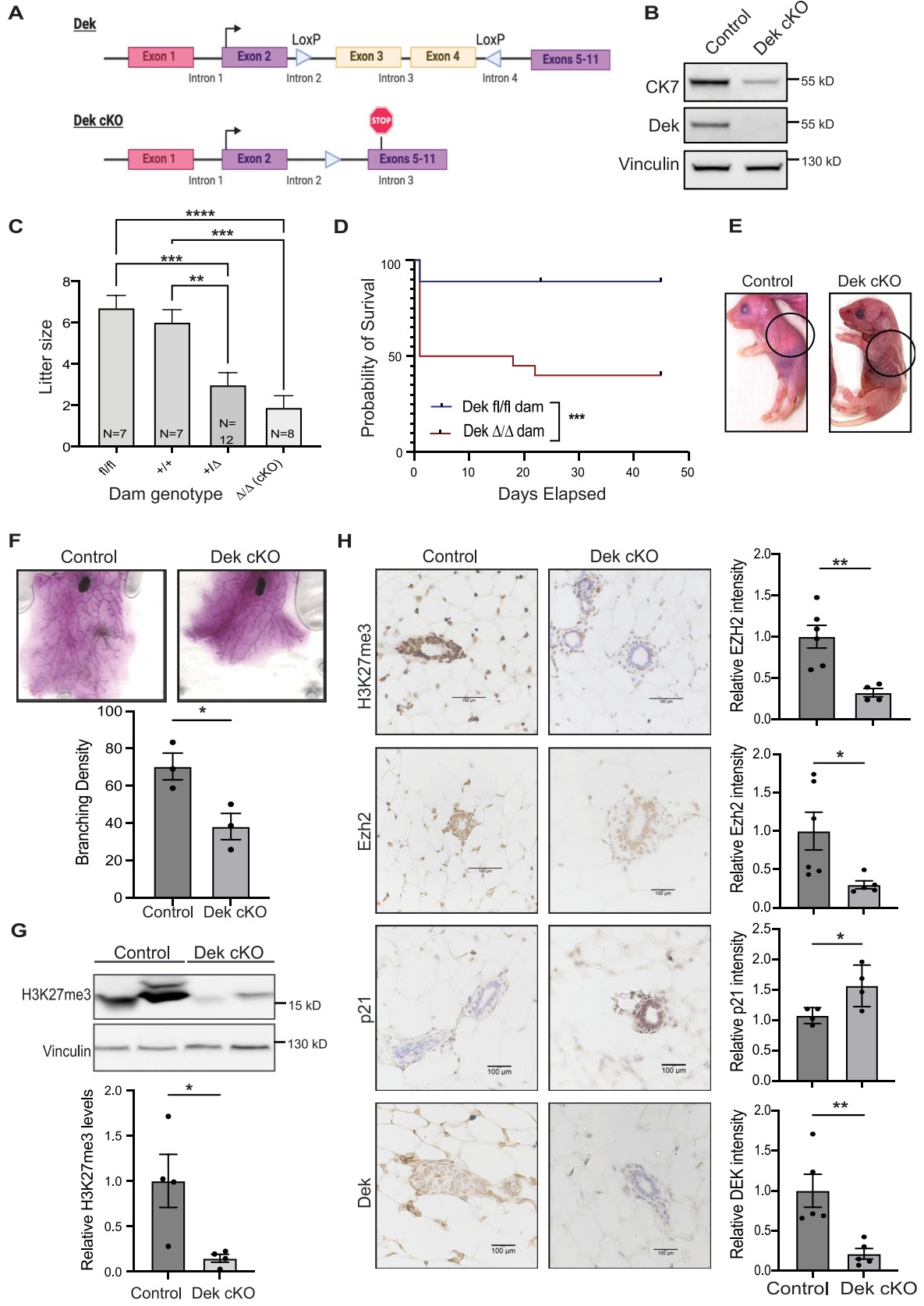

Importantly, genetic mouse models revealed that regulators of H3K27me3 levels are essential for maintaining the luminal alveolar progenitor cell population and proper alveologenesis. Pregnancy results in a redistribution of H3K27me3 marks across the genome in luminal progenitor cells and increased Ezh2 expression (52). Mice with mammary gland specific loss of *Ezh2* (MMTV-Cre/EZH2$^{f/f}$) have a 14-fold decrease in gland repopulating mammary stem and progenitor cells, reduced alveolar cell development, and lactation failure (52). On the contrary, Ezh2 over-expression in murine mammary glands results in hyperproliferation of luminal cells and delayed involution (53). These phenotypes closely resemble the ones we report here for *Dek* knockout and over-expression, respectively. Combined, it is evident that tightly regulated Ezh2 expression, and H3K27me3 levels, are necessary for proper mammary alveologenesis during pregnancy, subsequent lactation, and involution of the gland at weaning. Our data indicate that Dek is an important component of this epigenetic regulation of mammary gland development and alveologenesis. Future studies will be required to determine how Dek may be involved in PRC2 complex recruitment to promoters and/or methyltransferase activity and how this intersects with p53 transcriptional activity. Our work suggests that Dek may prevent the expression of p53 target genes, like *Cdkn1a*, and others through H3K27me3 epigenetic silencing of their promoters. This has significant implications for pathogenesis within the mammary gland, since p53 is a potent tumor suppressor, frequently mutated in breast cancers, and DEK and EZH2 are both highly expressed in breast cancer with oncogenic activity, particularly in triple negative breast cancers (17, 18, 51, 54, 55, 56). Although our Dek-OE transgenic mice did not form tumors through 15 mo of follow-up, the hyperplasia phenotype suggests that Dek may support tumor progression, rather than initiation. The use of these genetic mouse models of mammary-specific Dek expression will be useful tools for understanding the role of Dek in breast cancer pathogenesis and mammary gland development in vivo.

Based on gene ontology analysis, Dek also up-regulated the expression of multiple genes linked to the metabolism of RNA and proteins, with more specific cellular processes of nonsense-mediated decay, telomere maintenance, ribosomal RNA processing (data not shown, NES = 2.02, FDR $P$ = 6.1 × 10$^{-9}$), and

translation. This is in agreement with previous reports that DEK supports telomere integrity and maintenance by facilitating the deposition of histone H3.3 at telomeres (8). In addition, GSEA plots showed that genes related to oxidative phosphorylation, mitochondrial function, cholesterol/sterol metabolism, and fatty acid biosynthesis were all down-regulated in Dek over-expressing mammary glands. Metabolic dysfunction and telomere maintenance are hallmarks of tumorigenesis and, when combined with evidence of hyperplasia, support the interpretation that Dek over-expression may promote, but not initiate, tumorigenesis. Whereas we did not investigate the impact of Dek over-expression on translation and metabolic processes directly in this manuscript, it is certainly worth investigating in the future. Interestingly, these cellular processes are supported by previous work that defined the DEK interactome in HeLa cells. DEK interacting proteins such as IMPDH2, DDX21, and numerous proteins of the large ribosomal subunit (RPL protein family) also implicate DEK in metabolism of RNA (nucleotide synthesis), ribosome RNA synthesis, and translation (39). Increased DEK expression has also been linked to AKT pathway activation, which regulates translation and exit from quiescence (12, 51, 57). Matrka et al reported that DEK over-expression in human epithelial cells reprograms metabolism to support macromolecule synthesis (58). With exit from quiescence and activation of the cell cycle, it is essential that the cell up-regulates the production of nucleotides and amino acids to double their mass for eventual mitosis, thus metabolism and cell proliferation are intricately connected. More work is needed to understand how DEK expression impacts macromolecule synthesis and metabolic reprogramming to support cell proliferation.

Visually, whole mount analysis revealed Dek-induced hyperplasia was limited to alveolar bud-like structures and not ducts, and thus this alveolar expansion was detected as a bias toward increased expression of luminal alveolar cell markers using RNA-Seq. We observed that Dek induced the expression of genes associated with lactation, including milk proteins even in virgin mice, as well as luminal alveolar cell markers, and stem/progenitor cell markers. Furthermore, a newly created conditional knockout mouse presents with decreased expression of luminal marker CK7 and poor pup survival before weaning. We also observed striking loss of H3K27me3 levels in Dek-deficient mammary glands. However, it is

**Figure 7. Mammary glands from Dek knockout mice are deficient in H3K27me3 and do not support neonatal survival.**
**(A)** Graphical representation of the floxed Dek allele in a novel conditional Dek knockout mouse model. LoxP sites flank exons 3 and 4 which, when removed by Cre recombinase, create a premature stop codon in exon 5. Exon 1 (pink) of Dek is non-coding. **(B)** Dek flox mice were bred to CMV-Cre mice to create a whole-body knockout. Whole cell lysates from mammary glands collected from 5-wk-old virgin female DEK knockout mice show loss of Dek expression and lower levels of luminal marker cytokeratin 7 (CK7). Vinculin is used as a loading control. **(C)** Dek knockout females have smaller litter sizes than Dek WT and CMV-Cre$^-$/Dek$^{fl/fl}$ controls. N values represent the number of separate dams in each Dam genotype group. N = 7 for Dek$^{fl/fl}$ and Dek$^{+/+}$ controls, N = 12 for Dek$^{+/Δ}$ heterozygous dams, and N = 8 for Dek$^{Δ/Δ}$ knockout (Dek cKO) dams. Significance calculated with one-way ANOVA with dam genotype as the variable. **(D)** Kaplan-Meier survival curves reveal that half of pups born to Dek$^{Δ/Δ}$ knockout dams (N = 17 pups) die within 24 h with a few more not surviving until weaning at 28 d. Minimal loss of pups is observed in litters born to Dek$^{fl/fl}$ (N = 20 pups). Data were collected from litters born to at least three separate dams per dam genotype. Statistical significance is determined by log-rank test. ***$P$ < 0.001. **(E)** Photographs of pups born to Dek WT (left) and KO (right) dams. Pups born to Dek WT dams had an observable milk spot within 24 h. Deceased pups from Dek KO dams were dehydrated and did not have a milk spot. **(F)** Whole cell lysates from virgin adult Dek$^{Δ/Δ}$ mammary glands had significantly less H3K27me3 levels than mammary glands from adult Dek$^{fl/fl}$ mice. (Bottom) Densitometry quantification of western blot data from (F) as determined by Image J. n = 4 per genotype. **(G)** DEK cKO mammary glands are smaller and have decreased branching density compared with WT mammary glands, as depicted by whole mount preparations (top). (Bottom) Branching density of Dek WT versus cKO mammary glands, as determined by Sholl analysis, is graphed. N = 3/genotype. All mice were virgin adult females. **(H)** p21 expression is increased, whereas Ezh2 expression, and H3K27me3 levels are lower, in the mammary epithelium of adult virgin Dek$^{Δ/Δ}$ mice compared with Dek proficient controls as determined by immunohistochemistry. n = 4–6/group. Quantification of staining intensity are graphed on the right and presented as mean ± SEM. For all studies, statistical significance is determined by an unpaired *t* test. *$P$ < 0.05, **$P$ < 0.01, ***$P$ < 0.001, ****$P$ < 0.0001, ns = not significant.
Source data are available for this figure.

noted that one weakness of this CMV-Cre knockout model is that there is a constitutive loss of Dek, which limits the interpretation for mammary epithelial cell-specific Dek functions. Regardless, our findings are supported by scRNA-Seq data that revealed Dek expression is highest in mammary stem and progenitor cell populations in murine glands and that Dek and Ezh2 are co-expressed in these populations. Altogether, we are the first to report that Dek expression strongly supports mammary gland development, particularly as it relates to pregnancy, by promoting proliferation of stem and luminal progenitor cells. It is unclear how this impacts eventual lactation and the amount of milk produced to sustain pup viability in our mouse models, which is another limitation of this work. We have previously reported that Dek is an estrogen and progesterone target gene (16); therefore, it is possible that Dek facilitates epigenetic remodeling and transcriptional control of gene expression in response to elevated pregnancy hormones. It will be important to investigate this potential mechanism in future studies.

In summary, we report novel functions for Dek in promoting H3K27me3 epigenetic modifications through Ezh2 expression and interactions with the PRC2 complex in the mammary gland. We also are the first to report that Dek supports mammary gland development through cell cycle control and, likely, RNA and protein synthesis. These findings have significant implications for understanding normal mammary gland development and disease pathogenesis, such as breast cancer.

## Materials and Methods

### Mice

Bi-L-Dek mice were kindly donated by Susanne Wells (Cincinnati Children's Hospital Medical Center, Cincinnati, OH) (30) and MMTV-tTA mice were generously provided by Kay-Uwe Wagner (University of Nebraska Medical Center, now at Wayne State University) (32). Bi-transgenic mice were generated and maintained on an FVB/N background. Dek-OE mice were generated by continuously mating MMTV-tTA and Bi-L-DEK mice until homozygosity was achieved and copy number stabilized, as determined by quantitative PCR with genomic DNA. Handling of mice was performed with the approval of the Cincinnati Children's Institutional Animal Care and Use Committee and approved under protocol 2020-0037 and 2023-0043. All mice were housed in specific pathogen-free housing with ad libitum access to food, with or without doxycycline, and water with a 12-h light/12-h dark cycle. DOX control animals were continuously fed DOX chow from maternal consumption during gestation through adulthood until the time of tissue collection.

The *Dek* conditional knockout allele was generated using CRISPR technology to introduce 5′-and-3′ loxP sites sequentially to flank exons 3 and 4. The sgRNAs were selected according to the on- and off-target scores from the CRISPR design web tool CRISPOR (https://crispor.gi.ucsc.edu) (59). To insert the 5′ loxP site into intron 2, two sgRNAs (spacer sequences: GAGCTGTCAAGGT-TACAGTG and GGGTTCCTGTAGGACATAG) were transcribed in vitro

using the MEGAshorscript T7 kit (Thermo Fisher Scientific) and purified by the MEGAclear Kit (Thermo Fisher Scientific) and stored at −80°C. To prepare the injection mix, sgRNAs (25 ng/µl each) was mixed with Cas9 protein (IDT; 100 ng/µl) in 0.1X nuclease-free TE buffer and incubated at 37°C for 15 min to form the ribonucleoprotein complex (RNP). The donor oligo (Ultramer from IDT) with asymmetrical homologous arm design (60) and the loxP sequence was added to the RNP at the final concentration of 100 ng/µl. The zygotes from superovulated female mice on the C57BL6/N background (Taconic Biosciences) were injected with the mix via a piezo-driven cytoplasmic injection technique (61). Injected zygotes were subsequently transferred into the oviductal ampulla of pseudopregnant CD-1 females for birth. One pup with 5′ loxP in the litter was identified by PCR and Sanger sequencing and selected for breeding to homozygosity. To introduce the 3′ loxP site into intron 4, the zygotes from mice homozygous for 5′ loxP were collected and microinjected with Cas9 (100 ng/nl), sgRNA (spacer sequence: TTCCTCTAGACCCAGTTAGG; 75 ng/µl) and a donor oligo (100 ng/ul), followed by embryo transfer into pseudopregnant CD-1 females for birth. This resulted in two founder males carrying 5′-and-3′ loxP sites *in cis*, only one of which was able to sire progeny and establish the line. The line was back-crossed to C57Bl/6N for four generations to eliminate off-target genome editing and then re-bred to homozygosity, resulting in the conditional knockout Dekfl/fl line that was maintained separately for future use. To test for phenotypes caused by Dek loss, Dekfl/fl mice were bred to CMV-Cre mice on the C57Bl/6 background (strain #006054; Jackson Labs). Once the deletion allele achieved germline transmission, the CMV-Cre transgene was eliminated from the colony and the strain was maintained as a constitutive knockout line via mating of heterozygous (Dek+/Δ) males and females. Unless otherwise noted, adult virgin female mice were used for analyses.

Adult female C57Bl/6 mice underwent ovariectomy with isofluorane anesthesia. Pregnancy stage was determined by checking for plugs after placing a male and female in the same cage.

### Genotyping

Tail clips were digested with DirectPCR Lysis Reagent (Viagen Biotech) containing 0.6 mg/ml Proteinase K (Invitrogen) and protocol (need to read bottles/thermocycler). For PCR analysis, 1 µl of DNA was added to JumpStart Taq Ready mix from Invitrogen (product #P2893) using the manufacturer's specifications.

Transgenes were detected with the following primers:

Bi-L-Dek transgene (161 bp): (F): CAGTGACACAAGGGAAGGGTCAGA (R): AGCCACTGAACTGACCCACGT. Luciferase (167 bp): (F): AGTC-GATGTACACGTTCGTCAC (R): TGACGCAGGCAGTTCTATGC.

tTA (504 bp): (F): GCTGCTTAATGAGGTCGG (R): CTCTGCACCTTGGTGATC.

DEK (endogenous, 109 bp): (F) TCGAAATGCCATGTTAAAGAGCA (R): AAGGCTTTGGATGCATTAAGAAGT.

DEK conditional knockout models were genotyped with the following primers:

CMV-Cre: (F): GCGGTCTGGCAGTAAAAACTATC (R): GTGAAACAGC ATTGCTGTCACTT.

DEK (detects endogenous and floxed alleles): 5′loxP (F): AGT-GAAATTACTGGTCTGTGAAG (R): CTGAGTGGAACAGCTCCTATAG 3′loxP

(F): AGATGCTTCACCTTAGAGCTG (R): TCAGTTTGGAGCAAATTTCATTTCC DEK deletion allele: (5′loxP F): AGTGAAATTACTGGTCTGTGAAG (3′loxP R): TCAGTTTGGAGCAAATTTCATTTCC.

### In vivo imaging systems

Mice were intraperitoneally injected with 15 ng/g of luciferin and allowed to metabolize the luciferin for 5 min before sedation with inhaled isoflurane. Mice were imaged in the Perkin Elmer in vivo imaging systems Spectrum CT.

### Western blotting

Mammary gland tissues flash frozen in liquid nitrogen and stores at –80°C until ready to use. Tissues were homogenized using mortar and pestle with liquid nitrogen, transferred to 1.5 ml tube and weighed, with 5 ml of RIPA added for every 1 mg of tissue. RIPA consisted of 1% Triton, 24 mM sodium deoxycholate, 0.1% SDS, 0.16 M NaCl, 10 mM Tris pH 7.4, 5 mM EDTA, 10 mM $Na_3VO_4$, and 10 mM NaF, supplemented with a protease inhibitor cocktail (Sigma-Aldrich). Tissue was lysed on ice in a 4°C cold room for 4 h with vortexing every hour. Samples were sonicated 3x for 15 s with the amplitude set to 35. Tissue was incubated on ice an additional 20 min then cleared by centrifugation at 14,500$g$ for 25 min. Clearing by centrifugation and transfer to new tubes was performed three times to fully remove lipids. Protein quantifications were determined with a Bradford assay then 30 mg of protein was separated on a 8–16% or 12% pre-cast gel (Bio-Rad) using primary antibodies diluted 1:500–1:1,000 and secondary antibodies (1:2,500; Cytiva). See Table S1 for list of antibodies used in this work. Membranes were exposed to enhanced chemiluminescence reagents (Thermo Fisher Scientific) and imaged using the Bio-Rad Chemidoc. Densitometry was performed using Image J.

### Mammary gland whole mounts

Mice were euthanized and the fourth inguinal mammary glands were excised, spread on a cover slip and prepared for whole mount fixation and staining as previously described (62). Whole mounts were imaged via Nikon Eclipse Ci upright microscope with Nikon Digital Sight camera and Nikon NIS-Elements software or digitally imaged with a flatbed scanner for Sholl analysis as previously described (63).

### Immunohistochemistry

Tissues were fixed in 4% PFA, embedded in paraffin, sectioned at 4 $\mu$m thickness, and fixed onto slides. H&E-stained sections were analyzed for histopathology. Paraffin sections were deparaffinized in xylene and rehydrated for antigen retrieval in sodium citrate. Off-target background staining was reduced by pretreatment with 0.3% $H_2O_2$. Sections were then treated with the Mouse on Mouse peroxidase immunostaining kit (Vector Labs) with primary antibodies. Sections were treated with 1:500 dilutions of biotinylated secondary antibody and stained with diaminobenzidine (DAB) and counterstained with hematoxylin then mounted with Permount (Thermo Fisher Scientific). Images were captured at the indicated magnifications with a Nikon Eclipse Ci upright microscope with Nikon Digital Sight camera and Nikon NIS-Elements software. Image J color deconvolution was used to measure the staining intensity only within mammary epithelial cells from at least three fields of view per tissue from each mouse. Specifically, cross-sections of similarly sized ducts were outlined such that only the collective epithelial cells within that cross section were measured, removing background signal from the stromal cells. Only single cross-sections of ducts were analyzed to minimize the impact of epithelial hyperplasia in experimental mice compared with controls fed dox chow.

### Cell culture

#### Primary mouse mammary epithelial cell isolation and culture

Mammary glands were dissected and epithelial cells were isolated via differential centrifugation as previously described (64). Primary mammary epithelial cells were cultured in Epicult-B media with proliferation supplement (#05610; Stemcell Technologies) with 10 ng/ml EGF, 10 ng/ml bFGF, 4 mg/ml Heparin, and penicillin/streptomycin. For organoid cultures, 8-chambered slides were coated in 100% Matrigel, then primary mammary epithelial cells were trypsinized and resuspended in 2.5% Matrigel in 50:50 mix of Epicult-B and 3D assay media at a density of 5,000 cells per chamber as previously described (65). Palbociclib (1 mM) or diluent ($H_2O$), or GSK-126 (2 $\mu$M) or diluent (DMSO), were added when seeding cells in 3D culture and maintained through media changes every 3 d. Organoids were visualized on days 3 and 7. Cells isolated from control mice fed doxycycline chow were maintained in culture with 1 mg/ml doxycycline. Organoid volume was quantified using ImageJ to measure the diameter and, thus, to determine the radius. The volume of a sphere ($4/3\pi r^3$) was used to determine organoid volume, and multiple organoids were counted per condition from each of the mouse donors.

#### Human cells

MCF10A human immortalized mammary epithelial cells were acquired from ATCC and cultured as previously described in DMEM:F12 medium with 5% horse serum, EGF, hydrocortisone, cholera toxin, insulin, and penicillin-streptomycin (65). DEK over-expressing cells were created by transduction with a modified doxycycline-inducible pTRIPZ lentiviral vector into which full-length, untagged, DEK cDNA had been cloned using the SnaBI and EcoRI restriction sites. DEK knockdown cells were created by lentiviral transduction of pLKO.1-DEK shRNA (MISSION shRNA #TRCN0000013104; Sigma-Aldrich). Selection of transduced cells was performed with 2 $\mu$g/ml puromycin and DEK expression induced with 1 $\mu$g/ml doxycycline. HEK293 FLp-In T-Rex cells were cultured in DMEM with 10% FBS and 1% Pen–Strep 37°C, 95% humidity and 5% $CO_2$. Primary human mammary epithelial cells were purchased and cultured according to manufacturer's instructions (#7610; ScienCell Research Laboratories). These cells were transduced with pLKO.1-DEK shRNA and selected in 2 $\mu$g/ml puromycin.

### LAP-tag immunoprecipitation

This assay was performed as previously described and with manufacturer's instructions (Flp-Ub T-Rex Core kit, #K6500-01; Invitrogen) (39, 66). Briefly, HEK293 cells were transfected with

Lipofectamin 2000 and 2 µg of pcDNA5/FRT/TO DEK-His-GFP, or pcDNA5/FRT/TO His-GFP control, and 18 µg of pOG44. After analysis of expression (data not shown) HEK293 cells containing the inducible LAP-tag DEK fusions were transferred from six 10 cm plates into a spinner flask containing 250 ml DMEM/10% FBS in the presence of Pen/Strep then expanded to a total of 1 liter (roughly $10^6$ cells/ml). 24 h before immunoprecipitation, expression of the LAP-tag fusion was induced with 50 ng/ml tetracycline. Nuclear extracts were prepared as described ([39]) whereas 100 µl GFP-trap beads was washed three times with 1 ml wash buffer (10 mM Tris–HCl pH 7.5, 150 mM NaCl, 0.5 mM EDTA). The nuclear extract was added to the washed GFP-trap beads and incubated for 1 h on a rolling platform at 4°C. The beads were washed three times with 1 ml wash buffer and eluted three times with 100 µl 0.2 M glycine pH 2.5 and neutralized in 1 M Tris pH 10.4. The proteins were precipitated according to the Wessel/Flügge protocol ([67]) and resuspended in 100 µl 2% SDS. 2–4 ml of eluates were analyzed by SDS–PAGE electrophoresis, followed by either Coomassie or silver staining or immunoblotting with the indicated antibodies. All LAP IP buffers were freshly supplemented with 5 mM DTT, 0.1 mM PMSF, and 1× Complete Protease inhibitor mix. For immunoblotting, the following antibodies were used: DEK (A0315, 1:1,000; Abclonal), GFP (AF1483, 1:1,000; Beyotime), CK2 (A19683, 1:1,000; Abclonal), histone H3 (AF0009, 1:1,000; Beyotime), RBBP4 (A3645, 1:1,000; Abclonal), EED (A5371, 1:1,000; Abclonal), EZH2 (A16846, 1:1,000; Abclonal).

**Co-immunoprecipitation**

DEK expression in MCF10A pTRIPZ-DEK cells was induced by treating cells with 1 mg/ml doxycycline for 48 h before collection in IP lysis buffer (25 mM NaCl, 25 mM Tris–HCl, 1 mM EDTA, 1% Nonidet P-40, and 5% glycerol) supplemented with phosphatase and protease inhibitors as in the Western blotting section. Lysates were cleared with protein A/G beads and 1 mg of protein was incubated with 2 µg primary antibody (or 1:50 dilution if concentration was unknown) overnight with rotation at 4°C. Antibodies for DEK (#ab245429; Abcam) or SUZ12 (D39F6) (#3737; Cell Signaling Technology) were used for immunoprecipitation. Antibody-protein complexes were precipitated with protein A/G agarose beads for 3 h, washed, and removed from beads by boiling in 2x SDS–PAGE sample buffer with 6% β-mercaptoethanol then run on a 4–15% gradient gel for immunoblotting as described above. DEK (#610948, 1:1,000; BD Biosciences), SUZ12 (#3737, 1:1,000; Cell Signaling Technology), and EZH2 (#21800-1-AP, 1:2,000; Proteintech) primary antibodies were used to probe for co-immunoprecipitation.

**RNA sequencing**

Mammary glands were collected from nulliparous 60-wk old female mice, two Dek-OE and two Dek-OE on dox chow since birth. RNA was isolated from the mammary gland with Trizol and ~80–120 ng of RNA was amplified to generate cDNA. The initial amplification step for all samples was performed with the NuGEN Ovation RNA-Seq System v2 (NuGEN). The concentrations were measured using the Qubit dsDNA BR assay. cDNA size was determined by using a DNA1000 Chip. Libraries were then created for both samples. Specifically, the Nextera XT DNA Sample Preparation Kit (Illumina), was used to

create DNA library templates from the double-stranded cDNA. Concentrations were measured using the Qubit dsDNA HS assay. Then 1 ng of cDNA was suspended in Tagment DNA buffer. Tagmentation (fragmentation and tagging with the adapters) was performed with the Nextera enzyme (AmpliconTagment Mix; Illumina) by incubation at 55°C for 10 min. NT buffer was then added to neutralize the samples. Libraries were prepared by PCR with the Nextera PCR Master Mix and 2 Nextera Indexes(N7XX and N5XX) according to the following program: 1 cycle of 72°C for 3 min; 1 cycle of 98°C for 30 s; 12 cycles of 95°C for 10 s, 55°C for 30 s, and 72°C for 1 min; and 1 cycle of 72C for 5 min. Purified cDNA was captured on an Illumina flow cell for cluster generation. The size of the libraries for each sample was measured using the Agilent HS DNA chip (Agilent Genomics). Libraries were sequenced on the Illumina HiSeq 2500 following the manufacturer's protocol, with 75-bp paired-end sequencing and a coverage of 30 M reads.

Quantification of mRNA expression levels was based on the TopHat/Cufflinks pipeline of the CCHMC DNA sequencing and Genotyping Core. Paired-end reads were aligned to mouse genome build mm10 with STAR version 2.6.1. Principal component analysis highlighted potential batch effects resulting in the removal of one control sample and its paired Dek-OE sample for a final sample size of two per group. Transcripts per million were generated using the pseudo aligner Kallisto and processed through the R package NOISeq version 2.42.0 for batch correction. AltAnalyze version 2.1.4.4 (EnsMart72 database) was used for differential expression analysis on the batch-corrected data matrix (Table S2). The AltAnalyze supplied empirical Bayes moderated *t* test was performed followed by Benjamini–Hochberg adjustment for false discovery (FDR). Pathway enrichment was performed on genes that met a final threshold cutoff of a raw *P* value ≤ 0.05 and fold change ≥ 1.5 using the publicly available web-based tool, g:Profiler (https://biit.cs.ut.ee/gprofiler/gost accessed on 5 April 2023). g:Profiler results were visualized in node maps created in Cytoscape version 3.9.1. The volcano plot was created with the Enhanced Volcano package in R version 4.2.1 (https://github.com/kevinblighe/EnhancedVolcano) using all expressed genes. Genes with a raw *P*-value ≤ 0.05 upregulated ≥ linear 1.5-fold are indicated in red and genes significantly down-regulated ≤ linear -1.5-fold genes are indicated in blue. A subset of highly regulated or mammary gland-related genes is highlighted in boxes. The raw and processed RNA-sequencing data are available upon request.

**scRNA-seq analysis**

Data previously published by Saeki et al was obtained and the log normalized gene expression values from the six cell clusters was used for analysis ([37]). To identify genes co-expressed with *Dek*, we applied a rigorous filtering criterion: only genes expressed in at least 50% of the cells within each cluster were retained for co-expression analysis. For the co-expression analysis, cells expressing both *Dek* and the selected genes were included, whereas cells that did not express *Dek* or the gene of interest were excluded from the analysis. Pearson correlation analysis was then performed to assess the degree of linear relationship between *Dek* expression and the expression of each selected gene across the cell population. The resulting Pearson correlation coefficient for each

gene was used to rank the genes based on their strength of co-expression with *Dek*. GSEA was subsequently carried out using the ranked list of genes. The Pearson correlation coefficients served as the input ranking for the GSEA. We used both gene ontology (GO) and Hallmark gene sets for the enrichment analysis to identify biological pathways or processes associated with *Dek* co-expression.

## Statistics

Error bars depict standard errors of data collected from at least three mice. Significance was set at $P < 0.05$ (*$P < 0.05$, **$P < 0.01$). Log-rank test, $t$ test or ANOVA tests were used to test for significance using GraphPad Prism.

# Data Availability

All data are available in the main text or the supplementary materials. Mouse models and other unique resources are available upon request to the authors and may require a materials transfer agreement.

# Supplementary Information

# Acknowledgements

We thank Veterinary Services, Jeff Bailey and the Comprehensive Mouse and Cancer Core, the Gene Expression Core, and the DNA Sequencing Core facilities at Cincinnati Children's Hospital Medical Center. Thank you to Jonathan Cheek and Jordan Harris for technical assistance. We appreciate the expertise of Adam Lane, PhD (Cincinnati Children's Hospital) for feedback on statistical analyses and sample sizes calculations. Funding was provided by the following: Cancer Survivorship pilot grant (No award number) from the University of Cincinnati Cancer Center (LM Privette Vinnedge). Trustee Award (No award number) from Cincinnati Children's Hospital (LM Privette Vinnedge). National Institutes of Health grant T32CA117846-13 (ME Johnstone). Duke Kunshan University - Wang-Chai and Start-Up grant (F Kappes). National Institutes of Health grant R01CA239697 (SI Wells). National Institutes of Health grant R37CA218072 (LM Privette Vinnedge).

## Author Contributions

ME Johnstone: conceptualization, data curation, funding acquisition, methodology, and writing—review and editing.
AL Leck: data curation, formal analysis, validation, visualization, project administration, and writing—review and editing.
TE Lange: formal analysis, visualization, methodology, and writing—review and editing.
KE Wilcher: data curation.
MS Shephard: data curation.
A Paranjpe: data curation, formal analysis, visualization, and methodology.
S Schutte: data curation and formal analysis.
SI Wells: resources, supervision, funding acquisition, and writing—review and editing.
F Kappes: resources, data curation, formal analysis, funding acquisition, methodology, and writing—review and editing.
N Salomonis: methodology.
LM Privette Vinnedge: conceptualization, resources, data curation, software, formal analysis, supervision, funding acquisition, methodology, writing—original draft, and project administration.

## Conflict of Interest Statement

The authors declare that they have no conflict of interest.

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
