## [Reviewer comments · Life Science Alliance]

Life Science Alliance

DEK promotes mammary hyperplasia and is associated with H3K27me3 epigenetic modifications

Megan Johnstone, Ashley Leck, Taylor Lange, Katherine Wilcher, Miranda Shephard, Aditi Paranjpe, Sophia Schutte, Susanne Wells, Ferdinand Kappes, Nathan Salomonis, and Lisa Privette Vinnedge

DOI: <https://doi.org/10.26508/lsa.202503230>

Corresponding author(s): Lisa Privette Vinnedge, Cincinnati Children's Hospital Medical Center

Review Timeline:

Submission Date:	2025-01-18
Editorial Decision:	2025-01-21
Revision Received:	2025-04-24
Editorial Decision:	2025-05-31
Revision Received:	2025-06-08
Accepted:	2025-06-11

Scientific Editor: Tim Fessenden

Transaction Report:

Please note that the manuscript was reviewed at *Review Commons* and these reports were taken into account in the decision-making process at *Life Science Alliance*.

Review #1

1. Evidence, reproducibility and clarity:

Johnstone & Leck et al. report their findings on the DEK chromatin remodeler and its newly discovered role in the development of the mammary gland through the use of a mammary epithelium-specific Dek overexpression model (Dek-OE). Using immunohistochemistry (IHC) and whole mounts of mammary glands, they show that the Dek-OE model is characterized by epithelial hyperplasia in multiparous, 15-month-old females. Through performing and analyzing bulk RNA sequencing of whole mammary tissue, they find that overexpression of Dek is correlated with cell cycle entry and progression, and the expression of luminal alveolar and mammary progenitor genes. The deregulation of cell cycle inhibitors was confirmed through IHC and western blot. To further support the connection between Dek and the cell cycle, it was also shown that palbociclib treatment of mammary epithelial organoids derived from Dek-OE mice was able to rescue the hyperplastic phenotype. To validate their transcriptomic findings of increased expression of luminal progenitor genes, IHC and western blots for alveolar markers and milk proteins were performed. By performing ovariectomy and looking at DEK expression throughout the development of the mammary gland, it was also found that Dek expression was promoted by ovarian hormones. Analysis of single-cell data from a previously published single cell gene atlas of the mammary gland, the authors found that Dek expression heavily overlapped with mammary stem cells and luminal progenitor populations, and was heavily correlated with expression of PRC2 components. Using western blots and a GFP-trap assay, it was found that Dek overexpression leads to increased H3K27me3, and PRC2 components directly interact with DEK. Using a conditional knockout of Dek, the authors found that Dek loss leads to decreased expression of PRC2 components in mammary epithelial cells by IHC and a failure for dams to lactate efficiently. While the authors findings are novel, there are major points that need to be strengthened and elaborated for clarity.

****Major points:****

1. Several of the conclusions are made based on a limited number of replicates (often n=3) which is not a robust sample size to make a rigorous conclusion.
2. The main text for Figure 1C mentions repression of luciferase expression by doxycycline chow, however the figure does not show any discernable repression in the Dek-OE conditions.
3. To evaluate the impact of prolonged Dek overexpression on mammary epithelium in Figure 1G and 1H, the authors used multiparous females. One confounding factor with this experimental set up is the impact of previous pregnancies on the development of the mammary epithelium and in lowering tumorigenesis. Therefore, the impact of Dek on tumorigenesis cannot be determined in multiparous animals alone. To get a full picture, nulliparous animals should also be examined.
4. To elucidate the molecular underpinning of Dek-OE phenotypes, the authors performed bulk RNA sequencing in Figure 2. Similar to point 2 however, only multiparous animals were used. As it has been previously shown that pregnancy significantly impacts the transcriptome of mammary glands, the effects of Dek overexpression can't be generalized to mammary glands as a whole. To make it generalizable, nulliparous Dek-OE animals must also be characterized.
5. To validate findings from their transcriptomics work, the authors used IHC and western blots of candidate proteins that were found to be down regulated. In Figure 3A and 3C, the decrease in p21 protein levels through western blot seem much more modest than what the decrease seen in 3A would suggest.
6. In Figure 3G-3I, the authors test the CDK4/6 inhibitor palbociclib to establish a direct link between the phenotypes seen in Dek-OE and cell cycle progression in organoid culture. Have the authors verified these findings with treatment of Dek-OE mice with palbociclib? In addition, have the authors checked to see if palbociclib corrected any of the transcriptional features associated with the Dek-OE model found in their transcriptomics data? In addition, the authors claim that the effect is specific to Dek-OE organoids as the effects of palbociclib on growth are not seen in control organoids. However, the data on unperturbed growth of control cells are not seen. To determine the specificity of the effects of palbociclib on Dek-OE derived organoids, the authors must show a time course tracking the growth of organoids with and without palbociclib. Rather than conclude the effects of palbociclib being specific to Dek-OE organoids, the authors most likely wanted to conclude that the increased growth of Dek-OE organoids compared to control organoids is dependent on the increase in cell cycle factors. (The validity of this is also weird though because even if division and growth were triggered through other transcriptional changes they found, like increased metabolism, growth in that scenario would be stopped by palbo as well)

7. In the main text of Figure 4, the authors conclude that markers for luminal hormone sensing cells were unchanged in Dek-OE mammary glands, however the data to show this is not shown. This is problematic because the authors are directly drawing the conclusion that Dek-OE specifically upregulates luminal alveolar markers using this data.

8. In figure 7, the authors look at a conditional knockout of Dek and conclude that pup death in the knockout was due to insufficient milk production by dams. While the authors establish that H3K27me3 and Ezh2 expression are abrogated, morphological analysis of the ducts is missing and would present convincing data. For instance, in the Dek conditional knockout, are luminal alveolar cells unable to differentiate fully, or are there far fewer? Decreased levels of histone modifications does not tell you much about whether repressive chromatin has changed its landscape in Dek KO mice, which is actually what influences transcription.

****Minor points:****

All figures need some sort of reformatting. Several of the conclusions are made based on a limited number of replicates (often n=3) which is not a robust sample size to make a rigorous conclusion. Many figures have text that is stretched. Histology and whole mount images are missing scale bar. IHC quantifications are obscure - what is an optical density? how many animals were analyzed and how many fields of vision were captured? Figure 2F is absolutely impossible to understand. Neither figures nor legends disclose the number of animals or samples analyzed. The statistical test utilized across all figures is not appropriated. Fig5B GSEA plots are missing statistical significance, and without this information one cannot properly assess the relevance of the findings. Fig5C - how were co-expressed genes defined? is this just random genes that are expressed in cells that have higher levels of DEK? The term co-expressed suggests a specific type of analysis that would investigate linkage of expression between genes, which I don't think is the case here.

1. Throughout, it would be better to indicate the genotype of the "Control" animals on each figure so as the rigor of the experiment can be evaluated fully.
2. Standard nomenclature for gene names and protein names should be corrected throughout the text.
3. Similar to the point above, the use of Dek-OE to either refer to the mouse model or function as an acronym for "Dek overexpression" is inconsistent throughout the text.
4. In the main text for Figure 4I-J, the authors state that DEK was previously published as an Era target gene, however there is no citation to support this.
5. It is unclear what the conclusion drawn from the experiments shown in Figure 4G-H and Figure 4I-J mean with respect to the goal of Figure 4, which was to show that Dek-OE mice have an expanded luminal alveolar compartment.
6. Optical density was used to quantify IHC experiments, which was performed using color deconvolution in ImageJ. Something that is unclear is whether the authors are measuring density in the entire field of view, or if the authors are measuring optical density per cell. This has implications whether there are more cells expressing the protein of interest, or if the existing cells are expressing a higher level of the protein of interest.
7. In the main text for Figure 6D, the system being used to overexpress DEK protein is not described. It is not the same genetic system as is used in the Dek-OE mice, as doxycycline is inducing Dek expression.

2. Significance:

The role of Dek in tumorigenesis and in maintaining stem-like qualities in breast cancer cell lines have been previously reported. However, Dek has never been studied in the context of the normal mammary gland. The authors' work revealing the role of Dek in normal development of the mammary gland is significant as understanding it has the potential of revealing additional roles Dek may have as an oncogene in breast cancers.

Review #2

1. Evidence, reproducibility and clarity:

In the manuscript "The chromatin remodeler DEK promotes proliferation of mammary epithelium and is associated with H3K27me3 epigenetic modifications", Johnstone et al. investigate the proto-oncogene, DEK, in control of normal mammary gland development. The authors utilize transgenic mouse models, including conditional DEK-overexpression (OE) and DEK-knockout (KO) model, highlighting the role for DEK in control of mammary epithelial cell (MEC) proliferation and differentiation during pregnancy and lactation. Furthermore, the authors demonstrate DEK expression correlates with EZH2 and H3K27me3, which have previously been reported to control mammary gland lactation [Pal et al., Cell Reports, 2013].

Overall, this manuscript is interesting and well prepared. This group have previously established a role for DEK in breast cancer, however, the function of DEK in normal mammary gland development is unknown. Towards this goal, two novel mouse models were developed, conditional DEK-OE and DEK-KO. This manuscript would be substantially improved by formal investigation of virgin mammary gland development in these models, high-resolution analysis of the MEC subpopulations at different stages, and strengthening the mechanistic link between DEK and EZH2. The following are detailed major concerns.

1. This study would be improved by sharing important data including virgin mammary gland development in the DEK-OE and DEK-KO models (ductal growth and branching) and the expression of markers including ESR1, PGR, and ERBB2 (data not shown, page 8). Although there may be no differences, this is important data to share regarding the goal of this study. For example, in the DEK-OE model, data are only evaluated in the aged/multiparous stage and in the DEK-KO model, data are only evaluated during lactation. Furthermore, the DEK-KO model resembles germline DEK loss (under control of the CMV promoter), and there is limited validation of a MEC-intrinsic function.

2. Another major concern with this manuscript is the use of immunohistochemistry (IHC) and bulk mammary gland lysate western blots. IHC is non-quantitative, and the images are low resolution. For example, using IHC DEK expression is observed in all MECs (control and DEK-OE mice, Figure 1F), however, in the scRNAseq data DEK expression is confined to basal cells and a subset of stem/progenitor cells (Figure 5A). Furthermore, the hyperplasia in the DEK-OE model will bias bulk analysis (such as western blot and RNAseq) towards increased expression of MEC markers.

3. A third major concern is the mechanistic link between DEK and H3K27me3. Most of the data are correlative and rely on bulk analysis or IHC. For example, in the DEK-OE organoid model, is there an increase in H3K27me3. Additionally, in the DEK-OE organoids, can loss of EZH2 block the increased cell proliferation?

2. Significance:

Using genetic approaches in mice, this paper explores the role of the chromatin remodeler and oncogene DEK in the development of the normal mammary gland. This work will be of interest to researchers in the mammary development and breast cancer fields.

January 21, 2025

Re: Life Science Alliance manuscript #LSA-2025-03230-T

Dr. Lisa M Privette Vinnedge
Cincinnati Children's Hospital Medical Center and University of Cincinnati
Oncology
3333 Burnet Avenue
MLC 7018
Cincinnati, OH 45229

Dear Dr. Privette Vinnedge,

Thank you for submitting your manuscript entitled "DEK promotes mammary hyperplasia and is associated with H3K27me3 epigenetic modifications" to Life Science Alliance. We invite you to re-submit the manuscript, revised according to your Revision Plan.

Thank you for this interesting contribution to Life Science Alliance. We are looking forward to receiving your revised manuscript.

Sincerely,

Eric Sawey, PhD
Executive Editor
Life Science Alliance
<http://www.lsa-journal.org>

B. MANUSCRIPT ORGANIZATION AND FORMATTING:

Re: Resubmission letter for

Manuscript number: RC-2024-02726 → LSA-2025-03230-T

Corresponding author(s): Lisa Privette Vinnedge

Dear Editorial Members and Reviewers:

We thank the reviewers for their time and critiques, which have vastly improved the rigor of the manuscript. We hope the editors and reviewers find our revisions satisfactory and will reconsider this manuscript for publication in *Life Science Alliance*.

Both reviewers assumed much of the data shown was from pregnant or previously pregnant mice and requested a significant amount of preliminary data from virgin mice (R1 comment #3, #4, R2 comment #1). We regret not making it clear that most of the data was, in fact, from virgin mice, with a few exceptions. We have clarified the parity status in the figure legends and/or text for each experiment throughout the manuscript. Otherwise, we address the remaining reviewers' concerns below and edits within the manuscript are indicated with red text:

Reviewer 1:

1. **Major Comment 1:** "Several of the conclusions are made based on a limited number of replicates (often n=3) which is not a robust sample size to make a rigorous conclusion."
 - a. **We have consulted with a biostatistician (Adam Lane, PhD, now included in acknowledgements) regarding sample sizes. In nearly every subpanel, we have added new biological replicates based on his calculations. Most experiments include 5-8+ age- and parity-matched biological replicates.**
2. **Major Comment 2:** The main text for Figure 1C mentions repression of luciferase expression by doxycycline chow, however the figure does not show any discernable repression in the Dek-OE conditions.
 - a. **The mouse on the far left ("control") with no luciferase signal is the dox chow-repressed condition. We have revised the figure label to specify that "Control" is the "+dox condition." We now specify "+dox controls" throughout the text of the manuscript we have specified "+dox controls" instead of just "controls." Within each control/Dek-OE experiment, the mice are the same genotype and only differ in whether or not they consumed doxycycline containing chow.**
3. **Major Comments 3 & 4:** "To evaluate the impact of prolonged Dek overexpression on mammary epithelium in Figure 1G and 1H, the authors used multiparous females. One confounding factor with this experimental set up is the impact of previous pregnancies on the development of the mammary epithelium and in lowering

tumorigenesis. Therefore, the impact of Dek on tumorigenesis cannot be determined in multiparous animals alone. To get a full picture, nulliparous animals should also be examined.” AND “To elucidate the molecular underpinning of Dek-OE phenotypes, the authors performed bulk RNA sequencing in Figure 2. Similar to point 2 however, only multiparous animals were used. As it has been previously shown that pregnancy significantly impacts the transcriptome of mammary glands, the effects of Dek overexpression can't be generalized to mammary glands as a whole. To make it generalizable, nulliparous Dek-OE animals must also be characterized.”

a. **As mentioned in the general statement, we apologize for the lack of clarity in the experimental design and have corrected this. RNA-Seq, whole mounts (except Fig 4G), and all subsequent molecular validations were conducted on virgin mice. Data in Figures 1 and 2 were from aged virgins to quantify the impact of chronic Dek over-expression, while Figure 3 and 4A-F demonstrate similar transcriptional profiles in aged and young virgin females. The only exceptions are in Figure 4G-K, where we do detect endogenous Dek expression during pregnancy and the enhanced hyperplasia caused by Dek over-expressing during pregnancy in the transgenic model. We have corrected the confusing language about parity status and have specifically added the parity status to each experiment in the Results section and/or Figure Legends.**

4. Major Comment 5: To validate findings from their transcriptomics work, the authors used IHC and western blots of candidate proteins that were found to be down regulated. In Figure 3A and 3C, the decrease in p21 protein levels through western blot seem much more modest than what the decrease seen in 3A would suggest.

a. **We thank the reviewer for pointing this out. Added biological replicates (also a request of this reviewer), shows good alignment of p21 protein expression between western and IHC data.**

5. Major Comment 6 (beginning): In Figure 3G-3I, the authors test the CDK4/6 inhibitor palbociclib to establish a direct link between the phenotypes seen in Dek-OE and cell cycle progression in organoid culture. *Have the authors verified these findings with treatment of Dek-OE mice with palbociclib?....*

a. **Because the hyperplasia phenotype accumulates over the lifetime of the animal, the amount of treatment time required to abrogate the hyperplasia phenotype could be from days to weeks to months. For this reason, we believe it is outside the scope of this revision to assess the impact of palbociclib treatment in vivo for this manuscript. However, during the course of preparing a grant application on this project, we generated DEK deficient primary human mammary epithelial cells (MECs). The phenotypic results were so striking, we added them to this manuscript in Supplementary Figure 7. These results included increased p27, loss of Ki67+ cells, senescence morphology, and significant decreases in H3K27me3 deposition. We have placed this in the supplementary because**

the data is from only 1 donor. However, we feel it supports our conclusions that DEK promotes cell proliferation and cell cycle regulation through regulation of p27 expression and epigenetic mechanisms.

6. Major Comment 6 (middle of comment): ... In addition, have the authors checked to see if palbociclib corrected any of the transcriptional features associated with the Dek-OE model found in their transcriptomics data? In addition, the authors claim that the effect is specific to Dek-OE organoids as the effects of palbociclib on growth are not seen in control organoids. However, the data on unperturbed growth of control cells are not seen. To determine the specificity of the effects of palbociclib on Dek-OE derived organoids, the authors must show a time course tracking the growth of organoids with and without palbociclib.
 - a. **We appreciate this suggestion. In response, we repeated the experiment with palbociclib treatment of primary murine mammary organoids with palbociclib added to the organoids from the time of plating and for 7 days. We imaged and quantified organoid volumes on days 3, 5, and 7 but due to space constraints we only show days 3 and 7 in Figure 3G-H. While control organoids grow over time ($p=0.03$ by t-test). 2-way ANOVA revealed a time x group interaction ($F(3,10) = 8.475$, $p = 0.0042$) such that statistically significant values were only identified for the Dek-OE control group when compared to DekOE + palbociclib and both control groups on day 7. Therefore, this indicates the Dek-OE associated hyperplasia may depend on CDK4/6 activity.**

7. Major Comment 6 (end of comment): ... Rather than conclude the effects of palbociclib being specific to Dek-OE organoids, the authors most likely wanted to conclude that the increased growth of Dek-OE organoids compared to control organoids is dependent on the increase in cell cycle factors. (The validity of this is also weird though because even if division and growth were triggered through other transcriptional changes they found, like increased metabolism, growth in that scenario would be stopped by palbo as well)
 - a. **We have revised the text on page 8 to say, “We observed that the increased growth of Dek over-expressing organoids was dependent on the Dek-induced increase in CDK4/6, since palbociclib treatment resulted in smaller Dek over-expressing organoids that were comparable to organoids from +dox controls.” We would like to point out that with wild-type cells, organoid growth is naturally limited as cells within the center of the organoid undergo apoptosis and the empty lumen forms due to unfavorable environmental conditions. There is also less accessibility to growth factors to stimulate growth for receptors that are localized on the apical surface facing the lumen. Thus, it is not surprising to us that palbociclib has limited impact on wild-type organoid growth, since multiple mechanisms limit their growth.**

8. Major Comment 7: In the main text of Figure 4, the authors conclude that markers for luminal hormone sensing cells were unchanged in Dek-OE mammary glands, however the data to show this is not shown. This is problematic because the authors are directly drawing the conclusion that Dek-OE specifically upregulates luminal alveolar markers using this data.
- a. **We have revised the manuscript to include a new supplementary figure (now Fig S4) to include a western blot for HER2, PR, and ERα showing no significant differences in expression between Dek-OE glands and +dox controls. We also include a summary of RNA expression data from the bulk RNA-Seq experiment with markers of hormone-sensing and basal cell populations. This is limited by the small sample size (N=2) of the bulk RNA-Seq, but it is evident that some, but not all, luminal markers are higher in Dek over-expressing glands but the basal cell markers have more variability in their expression levels. Despite expression of basal markers trending higher in Dek-OE glands, the degree of variability leads to the conclusion that they are not significantly different at this time.**
9. Major comment 8: In figure 7, the authors look at a conditional knockout of Dek and conclude that pup death in the knockout was due to insufficient milk production by dams. While the authors establish that H3K27me3 and Ezh2 expression are abrogated, morphological analysis of the ducts is missing and would present convincing data. For instance, in the Dek conditional knockout, are luminal alveolar cells unable to differentiate fully, or are there far fewer? Decreased levels of histone modifications does not tell you much about whether repressive chromatin has changed its landscape in Dek KO mice, which is actually what influences transcription
- a. **We completely agree with the reviewer's critique and have done our best to address this concern. One limitation is that it is very difficult to acquire enough mice to complete these studies with rigorous sample sizes in the time frame given for revisions. With average litter sizes for heterozygous females being <=3 pups per litter (Fig 7C), we were not able to breed and age sufficient female mice for a comprehensive analysis. However, we did have 3 adult virgin females available for collection and whole mount analysis. Indeed, we found that mammary glands from Dek cKO mice had less branching density and were overall smaller than Dek-expressing controls (Fig 7F). In addition, we added histological evidence in new supplementary figure S8 that mammary glands from Dek deficient mice do not fully remodel during pregnancy and lactation (H&E) and that they produce less milk (Muc1+ staining in lumens). However, this data is placed in the supplementary figures due to limited sample size that prevented quantification and statistical analyses (see challenges explained above).**

Minor Points:

10. All figures need some sort of reformatting. Several of the conclusions are made based on a limited number of replicates (often n=3) which is not a robust sample size to make a rigorous conclusion. Many figures have text that is stretched. Histology and whole mount images are missing scale bar. IHC quantifications are obscure - what is an optical density? how many animals were analyzed and how many fields of vision were captured? Figure 2F is absolutely impossible to understand. Neither figures nor legends disclose the number of animals or samples analyzed. The statistical test utilized across all figures is not appropriated. Fig5B GSEA plots are missing statistical significance, and without this information one cannot properly assess the relevance of the findings. Fig5C - how were co-expressed genes defined? is this just random genes that are expressed in cells that have higher levels of DEK? The term co-expressed suggests a specific type of analysis that would investigate linkage of expression between genes, which i dont think is the case here.
- a. **As the reviewer mentioned in major comment #1, there was a concern with sample size, which we addressed above. We believe this concern about sample size was the rationale for the minor comment about “The statistical test utilized across all figures is not appropriate.” We have consulted with a biostatistician, Adam Lane PhD, who has confirmed that our statistical approaches were correct but confirmed the reviewer’s concerns regarding sample size. Thus, we have increased sample sizes and we hope this alleviates the reviewer’s concerns. Also, we have revised the text to include sample size information in figure legends and statistical significance information for GSEA plots in Fig 5.**
 - b. **The stretched text has been corrected. We have re-imaged IHC stained tissues to add size bars throughout. The reviewer asked about the number of fields of view for IHC quantification and this is now more clearly stated in the Methods, that at least 3 fields of view per tissue from each mouse. We have also added clarify text for the methods that states “Image J color deconvolution was utilized to measure the staining intensity only within mammary epithelial cells from at least 3 fields of view per tissue from each mouse. Specifically, cross-sections of similarly sized ducts were outlined such that only the collective epithelial cells within that cross section were measured, removing background signal from the stromal cells. Only single cross-sections of ducts were analyzed to minimize the impact of epithelial hyperplasia in experimental mice compared to controls fed dox chow.”**
 - c. **We have revised the text to add some clarifying sentences that we hope helps the reviewer better understand our methods for how genes co-expressed with Dek were identified in scRNA-Seq data. Specifically, the text now says:**

“To identify genes co-expressed with Dek, we applied a rigorous filtering criterion: only genes expressed in at least 50% of the cells within each cluster were retained for co-expression analysis. For the co-expression analysis, cells expressing both Dek and the selected genes were included, while cells that did not express Dek or the gene of interest were excluded from the analysis. Pearson correlation analysis was then performed to assess the degree of linear relationship between Dek expression and the expression of each selected gene across the cell population. The resulting Pearson correlation coefficient for each gene was used to rank the genes based on their strength of co-expression with Dek. Gene set enrichment analysis (GSEA) was subsequently carried out using the ranked list of genes. The Pearson correlation coefficients served as the input ranking for the GSEA. We used both Gene Ontology (GO) and Hallmark gene sets for the enrichment analysis to identify biological pathways or processes associated with Dek co-expression.”

d. Finally, we have added clarifying language in the Results section to describe Figure 2F, and to specify that it is a functional enrichment analysis for the upregulated genes identified in bulk RNA-Seq data.

11. Minor Point 1: Throughout, it would be better to indicate the genotype of the "Control" animals on each figure so as the rigor the experiment can be evaluated fully.
 - a. **We have revised the manuscript to better clarify that “controls” = “+dox chow” bitransgenics and have added text on page 5 to directly state this. We have also revised Fig 1C to specify that the mouse with no luciferase signal is the “+dox” control. Thus, all mice are the same genotype.**
12. Minor Point 2: Standard nomenclature for gene names and protein names should be corrected throughout the text.
 - a. **We have revised the text to confirm gene and protein names are correct. We have followed convention in using italics for gene names, non-italics for protein names, all capital letters for human genes/proteins (i.e.: DEK) and only first letter capitalization for non-human gene names (i.e.: Dek).**
13. Minor Point 3: Similar to the point above, the use of Dek-OE to either refer to the mouse model or function as an acronym for "Dek overexpression" is inconsistent throughout the text.
 - a. **We thank the reviewer for pointing out this inconsistency and we have revised the text so that the “-OE” notation is only used when discussing the mice and have changed to writing out “over-expression” for function.**
14. Minor Point 4: In the main text for Figure 4I-J, the authors state that DEK was previously published as an Erα target gene, however there is no citation to support this.
 - a. **We have revised the text to include this citation, which is:**

#16. Privette Vinnedge, L.M., et al., *The DEK Oncogene Is a Target of Steroid Hormone Receptor Signaling in Breast Cancer*. PLoS One, 2012. 7(10): p. e46985.

Thus, our work indicates that DEK is an ER α target gene, but there is not a feedback mechanism whereby DEK controls *Esr1* gene expression.

15. Minor Point 5: It is unclear what the conclusion drawn from the experiments shown in Figure 4G-H and Figure 4I-J mean with respect to the goal of Figure 4, which was to show that Dek-OE mice have an expanded luminal alveolar compartment.
 - a. **We have revised the text to better explain that we were investigating the impact of ovarian hormones and pregnancy on endogenous Dek expression in wild-type mice, since this information has not been previously reported and adds context to our study of the role of Dek in the mammary gland.**

16. Minor Point 6: Optical density was used to quantify IHC experiments, which was performed using color deconvolution in ImageJ. Something that is unclear is whether the authors are measuring density in the entire field of view, or if the authors are measuring optical density per cell. This has implications whether there are more cells expressing the protein of interest, or if the existing cells are expressing a higher level of the protein of interest.
 - a. **We have revised the text to include more information in the methods. The Methods now states:** “Image J color deconvolution was utilized to measure the staining intensity only within mammary epithelial cells from at least 3 fields of view from at least 3 different mice per group. Specifically, cross-sections of similarly sized ducts were outlined such that only the collective epithelial cells within that cross section were measured, removing background signal from the stromal cells. Only single cross-sections of ducts were analyzed to minimize the impact of epithelial hyperplasia in experimental mice compared to controls fed dox chow.”

17. Minor Point 7: In the main text for Figure 6D, the system being used to overexpress DEK protein is not described. It is not the same genetic system as is used in the Dek-OE mice, as doxycycline is inducing Dek expression.
 - a. **We have revised the figure 6 legend to specify** “DEK over-expression was accomplished with a dox-inducible pTRIPZ vector while DEK knockdown was accomplished with a pLKO.1 shRNA vector” **and we kindly point the reviewer to the Methods section (“human cell lines” subsection) as written in the first submission which included detailed information for the subcloning of DEK cDNA into the pTRIPZ vector.**

Reviewer 2

1. Comment 1: This study would be improved by sharing important data including virgin mammary gland development in the DEK-OE and DEK-KO models (ductal growth and branching) and the expression of markers including ESR1, PGR, and ERBB2 (data not shown, page 8). Although there may be no differences, this is important data to share regarding the goal of this study. For example, in the DEK-OE model, data are only evaluated in the aged/multiparous stage and in the DEK-KO model, data are only evaluated during lactation. Furthermore, the DEK-KO model resembles germline DEK loss (under control of the CMV promoter), and there is limited validation of a MEC-intrinsic function.
 - a. **We have revised the manuscript to include data on Esr1/ERa and Erbb2/Her2 by western blot in new Fig S4 as well as the bulk RNA-Seq mRNA levels (by FPKM) for select basal and luminal markers. The concern regarding parity was also mentioned by Reviewer 1 (major comments 3&4 above) that we believe was a misconception. Briefly we have clarified that nearly all data in the manuscript are from nulliparous (virgin) females unless otherwise stated, and we have revised the text throughout to more clearly state this fact. We have also revised the text to address the limitation of the CMV promoter. The Discussion section now states “However, it is noted that one weakness of this CMV-Cre knockout model, is that there is a constitutive loss of Dek, which limits the interpretation for mammary epithelial cell-specific Dek functions.”**

2. Comment 2: Another major concern with this manuscript is the use of immunohistochemistry (IHC) and bulk mammary gland lysate western blots. IHC is non-quantitative, and the images are low resolution. For example, using IHC DEK expression is observed in all MECs (control and DEK-OE mice, Figure 1F), however, in the scRNAseq data DEK expression is confined to basal cells and a subset of stem/progenitor cells (Figure 5A). Furthermore, the hyperplasia in the DEK-OE model will bias bulk analysis (such as western blot and RNAseq) towards increased expression of MEC markers.
 - a. **We have revised the text to point out that IHC images for Dek in control tissues show some cells have higher expression than others, which is what would be predicted by scRNA-Seq. The text now states on page 16 “The scRNA-Seq data suggests that Dek is more highly expressed in specific subpopulations of cells, and the variable intensity of immunohistochemical staining for Dek in epithelial cells within control mouse tissue supports this (see Fig 3I, 4I, 4K, and 7H).” Furthermore, on page 10 in the Result section we have revised the text to state “The mammary gland undergoes substantial hormone-induced remodeling across the murine lifespan and we wanted to determine if Dek expression varied during these remodeling processes. We show that Dek is highest during pregnancy and minimally expressed during lactation and involution (Fig 4K), and that Dek protein expression is not uniform across all epithelial cells in wild-type glands (Fig 3I, 4I-K). This suggests that certain epithelial subpopulations express more Dek than others.” We would**

also like to point out that, while endogenous Dek expression is limited to a subpopulation of cells, the MMTV promoter drives transgene expression in all epithelial cell subpopulations within the gland (Fig 1F and 3I, and Sakamoto, K. et al, *PLoS One*, 2012 reference #32). This is a limitation of the MMTV promoter. However, this also enriches our ability to see the transcriptional consequences of Dek expression in all mammary epithelial cells, which can provide insight into why its expression is needed in progenitor cells.

- b. We acknowledge that IHC and western blots are only semi-quantitative, which is why we attempt to perform both as orthogonal approaches or find additional ways to support our findings throughout the manuscript (i.e.: co-expression at the RNA level from gene expression databases or small molecule inhibitor treatment, etc). We also note that these methods are used to validate the quantitative method of RNA-Seq, and (often) validation of differentially expressed genes can be limited by antibody availability and the applications those antibodies are suitable for. However, we have specified in the methods that we used Image J and color deconvolution to specifically outline and assess expression in mammary epithelial cells by IHC, removing background staining from adipose cells and other stromal cells. Please see response #16 to reviewer 1's request for more information about how IHC staining intensity was quantified specifically for mammary epithelial cells.
- c. We also have revised the text to acknowledge that we knew the bulk RNA-Seq would be biased towards the hyperplastic cells. We wanted to take advantage of that bias to identify a gene signature that could be used to determine which cell type was leading to the hyperplasia phenotype. We used the differentially expressed genes to identify biomarkers for specific cell populations. On pages 6-7 the text now reads "We performed bulk RNA sequencing on whole mammary tissue from two +dox control and two Dek-OE adult virgin females at 15 months of age to discover molecular targets regulated by Dek over-expression and to reveal a gene signature that could help identify the expanded cell population(s) in hyperplastic glands." And "DEGs were plotted as a heatmap and ontologies for biomarkers of cell populations were defined to help identify the expanded cell population driving Dek-induced hyperplasia."

3. Comment 3: A third major concern is the mechanistic link between DEK and H3K27me3. Most of the data are correlative and rely on bulk analysis or IHC. For example, in the DEK-OE organoid model, is there an increase in H3K27me3. Additionally, in the DEK-OE organoids, can loss of EZH2 block the increased cell proliferation?

- a. **We agree with the reviewer's critique that we were missing a mechanistic or functional link between Dek and Ezh2 activity, and we appreciate their feedback. To this end, we have added an experiment, Fig 6I, in which we treat 3D organoids of cells harvested from +dox control and Dek-OE mice with GSK-126, an EZH2 methyltransferase small molecule inhibitor, or diluent as a control. While GSK-126 did result in a modest decrease in organoid volume for +dox controls, this was not statistically significant. However, statistically significant values were identified for the Dek-OE diluent group compared to both +dox controls (+/- GSK-126) and the Dek-OE + GSK126 group. This indicated that Dek-induced hyperplasia is dependent upon Ezh2 methyltransferase activity.**

Thank you for your time and your consideration of this revised manuscript.

Sincerely

Lisa M. Privette Vinnedge

May 31, 2025

RE: Life Science Alliance Manuscript #LSA-2025-03230-TR

Dr. Lisa M Privette Vinnedge
Cincinnati Children's Hospital Medical Center
Oncology
3333 Burnet Avenue
MLC 7018
Cincinnati, OH 45229

Dear Dr. Privette Vinnedge,

Thank you for submitting your revised manuscript entitled "DEK promotes mammary hyperplasia and is associated with H3K27me3 epigenetic modifications". As you will see, both reviewers commend the significant improvement in place in the revised work. Reviewer 2 requested refining some claims in light of data provided. While the bulk changes in H3K27me3 marks are consistent with epigenetic rewiring by DEK, we concur with this reviewer that these claims should reflect the fact that locus-specific epigenetic changes were not formally tested here (point 3). We invite you to either quantify DEK heterogeneity in Fig 4K or adjust this claim as noted by this reviewer (point 4). The remaining major and minor points noted by this reviewer are left to your discretion. We would be happy to publish your paper in Life Science Alliance pending these changes and final revisions necessary to meet our formatting guidelines.

- Please add the X and Bluesky handles of your host institute/organization, as well as your own and/or one of the authors, to our system.
- The titles in the system and the manuscript file must be consistent with each other.
- Please remove the "Teaser" label from the summary blurb.
- We need a clean manuscript file without text highlighting.
- Please be sure to mention all authors in the authors' contribution section in the manuscript text.
- The contributions selected for Susanne Wells do not qualify them for authorship. Please either update the contributions in our system and the Author Contributions section of the manuscript, or let us know if the author needs to be removed (and added potentially to the acknowledgment section).
- There are labels of A and B panels in the legend for Figure S1. However, these panels are not marked in the figure itself. Please correct the figure, legend, and call-outs accordingly.
- We encourage you to revise the figure legend for Figure 3 such that the figure panels are introduced in alphabetical order.
- Please add call-outs for Figure S4A-E to your main manuscript text.
- Please include molecular weight markers for all gel electrophoresis blots in figures 1, 3, 4, 6, 7, S4, and S7.
- Please ensure scale bars are visible on Figures 4K, S7, S8.

LSA now encourages authors to provide a 30-60 second video where the study is briefly explained. We will use these videos on social media to promote the published paper and the presenting author (for examples, see <https://docs.google.com/document/d/1-UWCfbE4pGcDdcgzcmiuJl2XMBJnxKYeqRvLLrLS08s/edit?usp=sharing>). Corresponding or first-authors are welcome to submit the video. Please submit only one video per manuscript. The video can be emailed to contact@life-science-alliance.org

A. FINAL FILES:

- An editable version of the final text (.DOC or .DOCX) is needed for copyediting (no PDFs).
- High-resolution figure, supplementary figure and video files uploaded as individual files: See our detailed guidelines for

preparing your production-ready images, <https://www.life-science-alliance.org/authors>

B. MANUSCRIPT ORGANIZATION AND FORMATTING:

Sincerely,

Reviewer #1 (Comments to the Authors (Required)):

The authors have addressed all of my concerns from the prior review.

Reviewer #2 (Comments to the Authors (Required)):

Johnstone & Leck et al report on novel roles of the chromatin remodeler DEK in mammary gland biology. They find that overexpression of Dek using the MMTV promoter results in hyperplasia through the downstream loss of CDK inhibitors, as well as the induction of a luminal progenitor-like gene expression profile. In addition, it was found that the overexpression of Dek results in an increase in overall H3K27me3, and inhibition of Ezh2 (a methyltransferase previously shown to facilitate trimethylation) in organoids was sufficient to correct abnormal growth. Further supporting their data, the authors show that a conditional Dek knockout results in decreased lactation and mammary gland expansion in response to pregnancy.

The revised manuscript addresses many of our previous concerns and makes the study more polished and interpretable. The authors clarified the genotype and parity status of the animals throughout the study, included biological replicates to support the significance of their findings, and provided more detail on their methodology for image and single-cell analyses. In addition, new experiments were added to provide mechanistic insight into how Dek functions with other methyltransferases. However, we believe there are still a few points that require revision.

Major points:

1. The authors state that Dek-OE is associated with a pro-proliferative gene signature, but the gene signature seen in Figure 2 seems more akin to just a luminal alveolar signature.
2. The main text may be misinterpreted to imply that DEK directly regulates cell cycle factors, particularly in line 197, which states, "The observed DEK-driven regulation of cell cycle effectors was sufficient to create a pro-proliferative phenotype." and throughout the discussion where the authors speculate about the role of Dek in regulating cell cycle components. We recommend revising the writing to more directly acknowledge that the effects of Dek over-expression may not be from direct regulation of cell-cycle effectors, as this is not shown, but from upstream pathways that result in a pro-proliferative state (Myc, Ras, Wnt).
3. The overall analysis of H3K27me3 levels is superficial, and given that localizing of such marks on the genome is of greater relevance than total levels, the provided data cannot conclude that correction of growth is mediated by gene repression.
4. On line 235, it is stated that: "This suggests that certain epithelial subpopulations express more Dek than others." Please include quantification of Dek protein expression in luminal and basal layers to support this conclusion, as it is not readily apparent in the images presented.

Minor points:

1. Please standardize the location and font size for scale bars.
2. Some figures still contain stretched/compressed text (2C y-axis title, 2D y-axis, plot titles in 5E)
3. 2F is still very confusing. It is our understanding that the main point is that the DEGs are enriched for cell cycle genes, but what do the connecting lines indicate? If that is the only point, a more interpretable GSEA plot would be better here.
4. Is there no statistically significant increase for any of the markers shown in Figure S4?
5. In figure S8, it would be useful to include a ducts/gland quantification to support your claims that Dek-KO dams have impaired alveolar expansion, as that is what it seems you're indicating with the black asterisk.
6. Please use a consistent style for your schematics for 1A and 7A.
7. In figure 4, in addition to including ovariectomy data, it may also be insightful to include data from mice implanted with estrogen/progesterone pellets to purely isolate the effects of estrogen and progesterone. However, we acknowledge that this may be outside the scope of revision.

Response to Reviewers

We thank the reviewers and editor for their constructive comments and suggestions and appreciate mention of significant improvements in the revised manuscripts. In response to remaining concerns, we have added GSEA plots in Figure 2 and added a new Figure S1. We also edited the text of the manuscript where indicated. We hope the manuscript is now acceptable for publication.

Major points:

1. The authors state that Dek-OE is associated with a pro-proliferative gene signature, but the gene signature seen in Figure 2 seems more akin to just a luminal alveolar signature.

One of the requests in the minor points (see below) was to include gene set enrichment analysis (GSEA) plots to clarify the functional processes associated with differentially expressed genes. While Fig2E highlights only the luminal alveolar signature that we chose selectively, it does not include cell cycle genes and other relevant gene signatures. There is a large cohort of differentially expressed genes related to cell cycle regulation, and other cellular/molecular processes, that we neglected to show. We have relegated the network map in Fig2F to the new Supplementary Figure 1C and instead added GSEA plots highlighting several categories of upregulated genes: cell cycle transition, DNA double-strand break repair, chromatin remodeling, p53 signaling, and mammary gland development (specifically alveolar cells, stem cells, and prolactin signaling, supporting the luminal alveolar genes highlighted in Fig2E). Additional signatures associated with translation, nonsense-mediated decay, and telomere maintenance are now shown in Supplemental Data (Fig S1A). GSEA plots for down-regulated genes were also added as requested (Fig S1B) and largely fall into a broad range of metabolism-related pathways.

2. The main text may be misinterpreted to imply that DEK directly regulates cell cycle factors, particularly in line 197, which states, "The observed DEK-driven regulation of cell cycle effectors was sufficient to create a pro-proliferative phenotype." and throughout the discussion where the authors speculate about the role of Dek in regulating cell cycle components. We recommend revising the writing to more directly acknowledge that the effects of Dek over-expression may not be from direct regulation of cell-cycle effectors, as this is not shown, but from upstream pathways that result in a pro-proliferative state (Myc, Ras, Wnt).

We completely agree with this concern and have eliminated any statements that could be misinterpreted as direct transcriptional regulation by Dek. As stated by the reviewers, neither the previous data nor the new GSEA plots distinguish between direct or indirect cell cycle gene regulation by Dek and Ezh2/H3K27me3. Upstream pathways inducing a pro-proliferative state may be responsible. To address this, we added the following text to the second paragraph of the discussion:

"However, it is important to note that epigenetic and transcriptional regulation of cell cycle genes by Dek/Ezh2/H3K27me3 in the mammary gland could be direct or indirect. Direct regulation might involve Dek cooperating with Ezh2 and the PRC2 complex to deposit the H3K27me3 mark directly on the promoters of quiescence-associated genes like *Cdkn1a* and *Cdkn1b*, leading to their transcriptional silencing and persistent proliferation via Cdk4/6 activity to drive hyperplasia. Indirect regulation might relate to prior work wherein DEK has been linked to several mitogenic signaling pathways including AKT/mTOR and Wnt signaling, which could subsequently upregulate cell cycle gene expression to promote a pro-proliferative state.^[12, 17, 50, 51] Future studies will seek to define direct or indirect modes of regulation by defining functionally relevant Dek/Ezh2/H3K27me3-dependent molecules and pathways.:

3. The overall analysis of H3K27me3 levels is superficial, and given that localizing of such marks on

the genome is of greater relevance than total levels, the provided data cannot conclude that correction of growth is mediated by gene repression.

The reviewer is correct and we have now clearly stated that our work promotes total H3K27me3 deposition but that its localization to distinct functionally relevant genomic loci is important to discern. Relevant ChIP-seq experiments are not feasible within the 1 week time frame of requested response, but they are now mentioned as an important future area of research. Specifically, we state in the second paragraph of the discussion:

“Here, we report for the first time that Dek is co-expressed with Ezh2 in murine and human mammary tissues, promotes the total deposition of the H3K27me3 epigenetic mark in mammary epithelial cells, and physically interacts with PRC2 complex members EED, RBBP4, and EZH2 in human cells. It will be necessary, in the future, to perform ChIP-Seq experiments to determine if the localization of these H3K27me3 marks facilitated by DEK are deposited at functionally relevant loci within the genome.”

4. On line 235, it is stated that: "This suggests that certain epithelial subpopulations express more Dek than others." Please include quantification of Dek protein expression in luminal and basal layers to support this conclusion, as it is not readily apparent in the images presented.

Because this was a consistent concern for reviewers, and only peripherally relevant to the overall importance of the manuscript, we have removed these statements from the results and the discussion, together with the corresponding Figure 4K.

Minor points:

1. Please standardize the location and font size for scale bars.

We have confirmed that all size bars on microscopy images are highly visible and 100 um in length. The Nikon software for our microscope was upgraded between revisions so there may be minor differences in text fonts

2. Some figures still contain stretched/compressed text (2C y-axis title, 2D y-axis, plot titles in 5E)

This issue is now fixed.

3. 2F is still very confusing. It is our understanding that the main point is that the DEGs are enriched for cell cycle genes, but what do the connecting lines indicate? If that is the only point, a more interpretable GSEA plot would be better here.

GSEA plots now replace the network map in Fig2F, and additional GSEA plots have been added to SuppFig1.

4. Is there no statistically significant increase for any of the markers shown in Figure S4?

This is now FigS5. While some markers are trending towards significance (e.g.: *Prlr*, *Ptn* and *Cited1*), likely based on sample size (FigS5A), none are statistically significant.

5. In figure S8, it would be useful to include a ducts/gland quantification to support your claims that

Dek-KO dams have impaired alveolar expansion, as that is what it seems you're indicating with the black asterisk.

This is now FigS9, and the figure legend was updated. The black asterisk indicates areas where adipocytes remained filled, a phenotype that is highly atypical for lactating glands. We refer to the editor to request additional data.

6. Please use a consistent style for your schematics for 1A and 7A.

We refer to the editor for a consistent style.

7. In figure 4, in addition to including ovariectomy data, it may also be insightful to include data from mice implanted with estrogen/progesterone pellets to purely isolate the effects of estrogen and progesterone. However, we acknowledge that this may be outside the scope of revision.

This would be an important complementary experiment to the ovariectomy data. However, as noted by the reviewers, and given the 1-week time frame for resubmission, it is beyond the scope of this manuscript.

Editor's requested revisions:

We have made all requested revisions provided by the editor, including a) addition of molecular weight size markers for western blots, b) ensuring highly visible scale bars in all immunohistochemistry images, c) matching manuscript titles in the online submission portal, and d) fixing discrepancies in the figure legends.

June 11, 2025

RE: Life Science Alliance Manuscript #LSA-2025-03230-TRR

Dr. Lisa M Privette Vinnedge
Cincinnati Children's Hospital Medical Center
Oncology
3333 Burnet Avenue
MLC 7018
Cincinnati, OH 45229

Dear Dr. Privette Vinnedge,

Thank you for submitting your Research Article entitled "DEK promotes mammary hyperplasia and is associated with H3K27me3 epigenetic modifications". It is a pleasure to let you know that your manuscript is now accepted for publication in Life Science Alliance. Congratulations on this interesting work.

DISTRIBUTION OF MATERIALS:

Again, congratulations on a very nice paper. I hope you found the review process to be constructive and are pleased with how the manuscript was handled editorially. We look forward to future exciting submissions from your lab.

Sincerely,
